# MorphAgent: Empowering Agents through Self-Evolving Profiles and Decentralized Collaboration

## Abstract

Large Language Model (LLM) based multi-agent systems (MAS) have shown promise in tackling complex tasks, but often rely on predefined roles and centralized coordination, limiting their adaptability to evolving challenges. This paper introduces MorphAgent, a novel framework for *decentralized* multi-agent collaboration that enables agents to *dynamically evolve their roles and capabilities*. Our approach employs self-evolving agent profiles, optimized through three key metrics, guiding agents in refining their individual expertise while maintaining complementary team dynamics. MorphAgent implements a two-phase process: a warm-up phase for initial profile optimization, followed by a task execution phase where agents continuously adapt their roles based on task feedback. Our experimental results show that MorphAgent outperforms traditional static-role MAS in terms of task performance and adaptability to changing requirements, paving the way for more robust and versatile multi-agent collaborative systems.

## 1 Introduction

The rapid advancement of Large Language Models (LLMs) (Achiam et al., 2023; Touvron et al., 2023b) has ushered in a new era of artificial intelligence, enabling the creation of sophisticated AI agents capable of tackling complex tasks across various domains (Nakajima, 2023; Torantulino, 2023). As these AI systems become more intricate, there is a growing need for effective collaboration mechanisms that allow multiple agents to work together. This collaborative approach, known as Multi-Agent Systems (MAS) (Han et al., 2024), has shown great promise in addressing challenges that are too complex or diverse for single-agent systems (Hong et al., 2024; Liu et al., 2023).

While existing MAS implementations have shown promising results, they often rely on predefined roles (Li et al., 2023), centralized coordination (Guo et al., 2024; Chen et al., 2024), or rigid organizational structures (Wang et al., 2024b; Hong et al., 2024). These approaches limit cooperative resilience within MAS (Chacon-Chamorro et al., 2024), which focuses on robustness and adaptability in dynamic, unpredictable environments. Figure 1 presents two examples to illustrate the real-world challenges with details elaborated below:

**Example 1.1** (Domain shift). Domain shift refers to a change in the characteristics or requirements of a task as it progresses through different phases or contexts, presenting new challenges and requiring different skill sets. For instance, a scientific research project could begin with literature review, move to experiment design, and conclude with result analysis and paper writing. These transitions demand a flexible and adaptive multi-agent system that can seamlessly adjust its collaborative strategies and agent roles as the task progresses.

**Example 1.2** (Node Failure). In real-world applications, the reliance on *centralized coordination* in many existing MAS approaches (Chen et al., 2024; Hong et al., 2024) introduces a potential single point of failure. Consequently, the single point of failure can lead to cascading failures in MAS, where the entire system collapses if the central coordinator becomes unavailable. This vulnerability highlights the need for more robust, decentralized approaches to multi-agent collaboration (Chacon-Chamorro et al., 2024).

In this paper, we address these above challenges through a fully decentralized approach, where agents autonomously adapt their roles and strategies. However, a naive implementation of fully decentralized MAS presents *inefficiencies for role assignment* since the agents require many interac-

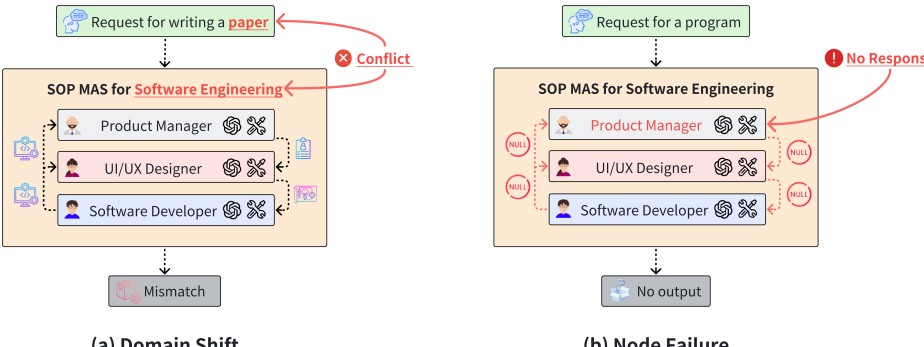

Figure 1: **Illustration of challenges in multi-agent systems (MAS) under domain shift and node failure scenarios.** (a) In the domain shift scenario, a user requests a paper, which is outside the expertise of the MAS that is optimized for software engineering tasks. The agents (Product Manager, UI/UX Designer, Software Developer) struggle to fulfill this request due to the mismatch in domain expertise, leading to a suboptimal result. (b) In the node failure scenario, a user requests a program, but due to the failure of the Product Manager node, the MAS experiences a cascading failure and is unable to complete the task.

tion rounds to converge on role assignments. Without a structured mechanism for role optimization, this process can lead to **suboptimal role distributions and slow adaptation to changing task requirements**, reducing the overall effectiveness of the system. This underscores the need for a more sophisticated approach to profile optimization in decentralized multi-agent systems.

To address above challenges, our design is motivated by the need to balance the benefits of decentralization with the efficiency of structured role optimization: the effective collaboration in a multi-agent system requires not just individual agent capability, but also a complementary distribution of roles that aligns with the task at hand. We introduce agent profiles as dynamic representations of evolving capabilities and responsibilities. Specifically, our approach employs quantitative metrics such as Role Clarity Score, Role Differentiation Score, and Task-Role Alignment Score to encourage self-optimization. By optimizing these metrics for agent profiles, our approach promotes clear role definition, diverse specialization, and task relevance, enhancing the system's adaptive capacity in dynamic environments.

Building on these insights, we propose a decentralized multi-agent system MORPHAGENT with adaptive profile optimization. Our method incorporates two key phases, namely, *a warm-up phase for initial role development*, and *a dynamic task execution phase*, thereby enables our system to achieve rapid initial role optimization while maintaining the adaptability necessary for long-term effectiveness. By allowing agents to autonomously adjust their profiles throughout the task execution, our method preserves the benefits of decentralization such as robustness and scalability while mitigating the potential inefficiencies of unstructured role assignment.

**Contributions**. We summarize the key contributions of this paper as follows:

- **Decentralized Collaboration Framework:** We introduce a novel, fully decentralized multi-agent collaboration framework designed to enhence system resilience and adaptability in complex tasks. By leveraging adaptive role optimization and fully decentralized coordination, our approach demonstrate robustness to node failures and adaptability to domain shifts.
- **Autonomous Collaboration Mechanism:** We develop an automatic collaboration system that does not depend on any critical agent or node. This approach distributes the process of decision-making and task execution across all agents, ensuring continued operation and performance even in the face of individual agent failures.
- **Adaptive Role Optimization:** We propose an adaptive role adjustment mechanism based on continuous profile optimization. By optimizing for role clarity, differentiation, and task alignment, our method enables a more flexible and robust form of collaboration. This mechanism allows agents to dynamically adjust their roles in response to dynamically task requirements and team composition changes.
- **Empirical Validation:** We provide comprehensive empirical evidence of our method's effectiveness through extensive experiments on standard benchmarks and custom-designed complex tasks. Our results show significant improvements over state-of-the-art baselines in terms of task performance, adaptability, and robustness to failures.

## 2 RELATED WORK

**LLM-based Multi-Agent Systems** The emergence of Large Language Models (LLMs) (Achiam et al., 2023; Touvron et al., 2023a) has led to LLM-based autonomous agents capable of tackling complex tasks, like BabyAGI (Nakajima, 2023) and AutoGPT (Torantulino, 2023). However, single LLM agents often struggle with cooperative work, such as software engineering (Jimenez et al., 2024). To address these limitations, recent study have proposed LLM-based multi-agent systems (MAS) (Han et al., 2024; Zhou et al., 2023), where multiple AI agents collaborate on complex tasks. Current approaches often rely on predefined roles, centralized coordination, or rigid organizational structures. CAMEL (Li et al., 2023) and ChatEval (Chan et al., 2023) employ agents with predened roles through role-playing, but struggle to adapt to tasks requiring unforeseen skills. MegaAgent (Wang et al., 2024b) introduces autonomous task splitting with centralized coordination, but this approach can create bottlenecks in large-scale systems and be damaged by single points of failure in real-world environments.. MetaGPT (Hong et al., 2024) implements human workflow in rigid organizational structures, showing improvements in code-generation but lacking generalization to other domains. Our work addresses these limitations by initializing agents homogeneously without predefined roles or structures, allowing them to naturally develop cooperation and specialization through interaction.

**Organization Optimization for MAS** Recent research in LLM-based Multi-Agent Systems (MAS) has focused on optimizing organizational structures (Guo et al., 2024; Zhuge et al., 2024) and enhancing agent performance (Zhang et al., 2024) to reduce communication costs and increase team efciency. Approaches like AgentVerse (Chen et al., 2024), Criticize-Reflect (Guo et al., 2024) and MegaAgent (Wang et al., 2024b) rely on centralized mechanisms, where a single role or a subset of agents monitor and evaluate the system's overall trajectory. While effective in certain scenarios, these centralized methods may face scalability issues and potential bottlenecks in large-scale MAS. Our research proposes a decentralized approach, leveraging LLM-based agents' self-reflection capabilities (Madaan et al., 2023; Shinn et al., 2023; Renze & Guven, 2024). Agents dynamically adjust their responsibilities based on context, enabling better scalability and mitigating context overload risks.

**Standard Operating Procedure based MAS** Another signicant strand of research has explored more structured and controlled methodologies in LLM-based multi-agent systems. Standard Operating Procedure (SOP) based approaches like AgentCoder (Huang et al., 2023) and MetaGPT (Hong et al., 2024) have shown performance gains through standardized pipelines. GPTSwarm (Zhuge et al., 2024) extends this by conceptualizing agents as subnets of action nodes. While effective for specific tasks, these approaches lack flexibility in dynamic environments. Our framework combines multi-agent collaboration with autonomous planning capabilities of advanced LLM-based agents (Huang et al., 2022; Guan et al., 2023; Wang et al., 2023). Instead of rigid SOPs, it enables dynamic development of collaborative strategies and efficient role adaptation, enhancing overall performance and robustness.

## 3 AUTONOMOUS COLLABORATION FOR FULLY DECENTRALIZED MAS

In this section, we present a novel decentralized multi-agent system which incorporated with a dynamic profile-based collaboration mechanism. Our key innovation lies in a two-phase process, as detailed in Appendix A, Algorithm 1: (1) *a warm-up phase* that optimizes agent profiles, and (2) *a task execution phase* where agents iteratively observe the environment, take actions, and update their profiles based on results and changes in the task state. To begin, we provide an overview of our multi-agent cooperation process in Section 3.1, followed by a detailed description of our dynamic profile-based collaboration mechanism in Section 3.2. Finally, we present the collaborative problem-solving process of our MAS in Section 3.3.

### 3.1 SYSTEM OVERVIEW

Complex tasks require adaptive problem-solving approaches that go beyond traditional centralized systems, which are vulnerable to bottlenecks and failure points. In centralized systems, a single coordinator is responsible for breaking down tasks and assigning them to individual agents. However, if the coordinator fails or communication is disrupted, the entire system risks collapsing. Additionally,

fixed roles within such systems prevent agents from adjusting to shifting task requirements, resulting in inefficiencies.

To develop a novel decentralized system, where multiple autonomous agents collaborate on complex tasks without predefined roles or centralized coordination, we introduce a fully decentralized multi-agent system, MORPHAGENT comprising three key components—namely, (i) Autonomous Agent, (ii) Auxiliary Agent, and (iii) Environment—each playing a crucial role in enabling autonomous, decentralized collaboration, as detailed in the following subsections.

**(i) Autonomous Agent.** The autonomous agent is built upon a LLM, e.g., GPT-4 (Achiam et al., 2023), which provides it with advanced reasoning and language understanding capabilities. It is primarily composed of a dynamic profile and utilizes the ReAct framework (Yao et al., 2022) combined with Reflexion (Shinn et al., 2023) to guide its behavior. The dynamic profile, which defines each agents roles and responsibilities, is central to how agents update and adapt, and will be discussed in detail in Section 3.2. Here, we focus on the operational mechanics of the agent.

Each agent follows the OBSERVE-THINK-ACT cycle, where it first gathers information from the environment and other agents in the OBSERVE phase. In the THINK phase, it uses its reasoning abilities to process the information and plan its next move as presented in Algorithm 1's Line 13. Finally, in the ACT phase, the agent decides to either EXECUTE or SKIP a task based on its current capabilities and the tasks requirements. After executing a task, the agent reflects on the outcomes, incorporating feedback into its memory for future decisions. This process is detailed in Algorithm 1's Line 18. The continuous cycle enables agents to adapt over time, refine their strategies based on feedback, and collaborate more effectively to address the specific problem at hand.

**(ii) Auxiliary Agent.** The auxiliary agent serves as a middleware component that facilitates inter-action between autonomous agents and the environment, without directly contributing to problem-solving. This design allows for efficient management of agent interactions and environmental feed-back while maintaining the decentralized nature of our system. By handling technical details, the auxiliary agent allows the autonomous agents to focus on high-level problem-solving strategies.

The main function of the auxiliary agent is to translate agent decisions into environmental operations and relays feedback from the environment back to the agents. For example, when an agent decides to execute Python code, the auxiliary agent runs the code in the environment, captures the output or errors from the environmental feedback, and provides this feedback to the multi-agent system. Thus the auxiliary agent ensures that autonomous agents can concentrate on strategic problem-solving while it manages execution and feedback processes.

**(iii) Environment.** The environment receives the instructions from users, processes them into messages, and the auxiliary agent is responsible for collecting and forwarding these messages within the environment to autonomous agents. Additionally, the environment is equipped with a mechanism to evaluate agent profiles, providing feedback that agents can use to refine and adjust their roles and strategies. This process ensures that agents continuously improve their task alignment and specialization, leading to more efficient collaboration and better overall system performance as task conditions evolve.

## 3.2 DYNAMIC PROFILE-BASED COLLABORATION MECHANISM

Traditional multi-agent systems (MAS) (Hong et al., 2024; Li et al., 2023) with *predefined roles* and *static structures* struggle to adapt to domain shift, leading to suboptimal performance when faced with unexpected changes or new challenges. To address this, we introduce a dynamic profile mechanism, as prensented in Algorithm 1's Line 4 and 22, enabling agents to continuously adjust their roles and skills based on interaction and task requirements. We begin with the concept of agent profiles, followed by the key metrics used to evaluate and optimize these profiles.

**Profile.** The concept of agent profiles encapsulates an agent's evolving role, capabilities, and exper-tise, serving as a dynamic representation of the agent's identity within the multi-agent system (Wang et al., 2024a). Profiles play a crucial role in MAS by facilitating effective task allocation, fostering collaboration, and enabling adaptive behavior (Sun et al., 2024). Unlike static role assignments, dynamic profiles allow agents to continuously refine their capabilities, leading to more flexible col-laboration and efficient problem-solving.

**Metrics for Profile Evaluation and Optimization.** In dynamic and complex environments, role ambiguity often significantly impair an agenet's effectiveness and efficiency, as agents struggle to understand their responsibilities and how they contribute to the overall task. Simultaneously, a lack of diversity may limit the system's ability to tackle multifaceted problems. For example, in a software development MAS, having too many agents specialized in front-end development but none in back-end or security could lead to an imbalanced and ineffective team. Furthermore, misalignment between agent capabilities and current task requirements can potentially result in ineffectiveness, especially when facing domain shift.

To address these issues and foster effective collaborationm, we propose three key metrics to assess and dynamically optimize each agent's profile: Role Clarity Score (RCS), Role Differentiation Score (RDS), and Task-Role Alignment Score (TRAS) as following definitions. These metrics guide the optimization process, ensuring that each profile accurately reflects the agent's role and capabilities (RCS), promotes diversity in the team (RDS), and aligns with the task requirements (TRAS).

**Definition 3.1** (Role Clarity Score (RCS)). For an agent $a \in \mathcal{A}$ with profile $p \in \mathcal{S}$, where $\mathcal{S}$ is the set of all possible profile strings, RCS considers the syntactic structure, lexical diversity, and skill relevance of the profile, which can be defined as:
$$\text{RCS}(a) = \beta_1 \cdot \text{DEP}(p) + \beta_2 \cdot \text{ENT}(p) + \beta_3 \cdot \text{SKILL}(p),$$
where $\beta_1 + \beta_2 + \beta_3 = 1$, and

- DEP $: \mathcal{S} \rightarrow [0,1]$ is the dependency score, measuring syntactic complexity. This builds on established principles in dependency parsing (Kübler et al., 2009) and syntactic role analysis (Jurafsky, 2000). It captures the structural depth and richness of the profile: $\text{DEP}(p) = h_1 \left( \frac{1}{|\mathcal{D}(p)|} \sum_{t \in \mathcal{D}(p)} |\mathcal{ST}(t)| \right)$, where $\mathcal{D}(p)$ is the set of tokens in $p$ involved in key dependency relations (e.g., subject, object), $\mathcal{ST}(t)$ is the subtree of token $t$, and $h_1 : \mathbb{R}_+ \rightarrow [0,1]$ is a normalizing function capturing syntactic complexity. Higher DEP scores indicate more detailed, complex profiles.

- ENT $: \mathcal{S} \rightarrow [0,1]$ is the entropy score, quantifying lexical diversity, defined as: $\text{ENT}(p) = h_2 \left( -\sum_{w \in \mathcal{W}(p)} \frac{f(w)}{|\mathcal{W}(p)|} \log_2 \left( \frac{f(w)}{|\mathcal{W}(p)|} \right) \right)$, where $\mathcal{W}(p)$ is the set of unique words in $p$, $f(w)$ is the frequency of word $w$ in $p$, and $h_2 : \mathbb{R}_+ \rightarrow [0,1]$ is a normalizing function. Higher ENT scores indicate diverse, less repetitive language.

- SKILL $: \mathcal{S} \rightarrow [0,1]$ is the skill score, measuring relevance to skill-related concepts, computed as: $\text{SKILL}(p) = \frac{\gamma}{|\mathcal{T}(p)|} \sum_{t \in \mathcal{T}(p)} \frac{e(t) \cdot e(s)}{\|e(t)\| \|e(s)\|} + (1 - \gamma) \frac{|\mathcal{PS}(p)|}{|\mathcal{T}(p)|}$, where $s$ is the skill prototype, a vector capturing the essence of skill-related concepts, defined as the average embedding of terms like "skill", "expertise", and "competence". $e(\cdot)$ is a word embedding function, $\mathcal{T}(p)$ is the set of tokens in $p$, and $\mathcal{PS}(p)$ is the set of potential skill tokens, identified through syntactic and semantic criteria, including similarity to $s$ and dependency relations (e.g., PROPN, NOUN in compound relations). Higher SKILL scores indicate stronger alignment with relevant skills.

**Remarks**: The RCS is motivated by the observation that well-defined roles in professional contexts exhibit distinct linguistic patterns: (a) they tend to have richer syntactic structures (captured by dependency scores), (b) more diverse and specific vocabulary (measured by entropy), and (c) clear skill specifications (quantified by skill relevance). By capturing these linguistic patterns, the RCS provides a measure of how well-defined and understandable an agent's role is. However, it doesn't account for the relationships between different agents' roles or their relevance to a specific task. To address these aspects, we introduce the following two metrics that consider a given task.

Given a task, MAS should exhibit a balance of diverse yet complementary roles. Thus, we introduce the Role Differentiation Score as follows:

**Definition 3.2** (Role Differentiation Score (RDS)). Let $\mathcal{A} = \{a_1, \ldots, a_n\}$ be a set of $n$ agents, with profiles $\mathcal{P} = \{p_1, \ldots, p_n\}$. The RDS of $\mathcal{A}$ measures the average dissimilarity between agent profiles, which can be defined as:
$$\text{RDS} = h_3 \left( \frac{2}{n(n-1)} \sum_{1 \leq i < j \leq n} d(a_i, a_j) \right),$$
where $d(a_i, a_j) = 1 - \frac{e(p_i) \cdot e(p_j)}{\|e(p_i)\| \|e(p_j)\|}$ is the dissimilarity between agents $a_i$ and $a_j$ measured by the embeddings of their profiles $p_i$ and $p_j$, and $h_3$ is a sigmoid function to normalize the score.

**Remarks**: The RDS is motivated by a fundamental principle in multi-agent systems: effective teams require *distinct* yet *complementary* roles to efficiently accomplish complex tasks. When roles are too similar, it leads to redundancy and potential resource inefficiency; when they're too different, it might create gaps in capability coverage. RDS quantifies this balance through pairwise profile dissimilarity measurements, capturing the degree of role specialization across the team. However, high role differentiation alone doesn't guarantee task-appropriate specialization, e.g., in a software development task, having agents with completely unrelated skills like a programmer and a chef would result in a high RDS but poor task performance. This motivates our next metric focused on Task-Role Alignment.

We introduce the Task-Role Alignment Score (TRAS) to ensure that the agents' roles are not only differentiated but also relevant to the given task:

**Definition 3.3** (Task-Role Alignment Score (TRAS)). Given a task $T \in \mathcal{T}$ and a set of agent profiles $\mathcal{P} = \{p_1, \ldots, p_n\}$, TRAS investigates how well the agents' roles align with the task requirements, which is defined as:

$$\text{TRAS} = \alpha \cdot S_{\text{sim}}(T, \mathcal{P}) + (1 - \alpha) \cdot S_{\text{cap}}(T, \mathcal{P}),$$

where:

- **Semantic Similarity** ($S_{\text{sim}}$): $S_{\text{sim}}(T, \mathcal{P}) = \frac{1}{n} \sum_{i=1}^{n} \frac{e(T) \cdot e(p_i)}{\|e(T)\| \|e(p_i)\|}$, where $e(T)$ and $e(p_i)$ are vector representations of the task and the $i$-th agent profile, respectively, obtained via a pretrained language model. These vectors capture the semantic proximity between task descriptions and agent profiles.
- **Capability Compatibility** ($S_{\text{cap}}$): $S_{\text{cap}}(T, \mathcal{P}) = 1 - \left| C_T(T) - \frac{1}{n} \sum_{i=1}^{n} C_A(p_i) \right|$, where $C_T(T)$ assesses task complexity and $C_A(p_i)$ evaluates the capability of an agent.
  $C_T(T)$ is defined as: $C_T(T) = \frac{1}{2} \left( 1 + \cos(\mathbf{v}_T, \mathbf{v}_{\text{complex}}) - \cos(\mathbf{v}_T, \mathbf{v}_{\text{simple}}) \right)$, where $\mathbf{v}_T$ is the vector representation of the task, and $\mathbf{v}_{\text{complex}}, \mathbf{v}_{\text{simple}}$ are vector representations of predefined complexity and simplicity indicators. Specifically, $\mathbf{v}_{\text{complex}}$ includes terms like "complex" for technical challenges, and "challenging" for task difficulty. Conversely, $\mathbf{v}_{\text{simple}}$ focuses on simplicity indicators like "basic" for scope and "routine", "standard" for effort.
  $C_A(p_i)$ evaluates the capability of the $i$-th agent profile as: $C_A(p_i) = \frac{1}{2} \left( 1 + \cos(\mathbf{v}_{p_i}, \mathbf{v}_{\text{capable}}) - \cos(\mathbf{v}_{p_i}, \mathbf{v}_{\text{limited}}) \right)$, where $\mathbf{v}_{p_i}$ represents the agent profile, while $\mathbf{v}_{\text{capable}}$ and $\mathbf{v}_{\text{limited}}$ capture capability and limitation indicators, respectively. $\mathbf{v}_{\text{capable}}$ includes terms like "expert", "experienced", and "certified", while $\mathbf{v}_{\text{limited}}$ covers "beginner", "junior", "novice", and similar terms reflecting entry-level abilities.

**Remarks**: The TRAS definition is motivated by the observation that effective multi-agent systems require roles that collectively match task requirements. This alignment has two critical dimensions: (a) semantic relevance (whether the roles' descriptions match the task's domain and requirements), and (b) capability compatibility (whether the team's skill levels match the task's complexity). TRAS captures these dimensions through $S_{\text{sim}}$, which measures *semantic proximity between role and task descriptions* using vector representations, and $S_{\text{cap}}$, which evaluates the *match between task complexity and team capabilities* using carefully defined indicator terms. By combining these complementary measures, TRAS provides a comprehensive assessment of how well-suited a team's role configuration is for a given task.

These three scores provide a comprehensive evaluation of the agents' roles, their differentiation, and their alignment with the given task. In Appendix B, we outlines the dynamic process of profile optimization, where these metrics are iteratively refined to enhance agent collaboration and task performance. We also provide a detailed implementation of these metrics in Appendix C. Additionally, Appendix D present quantitative examples evaluating these metrics on various agent teams for specific tasks, demonstrating how the metrics function in practice.

To optimize the agent profiles, we leverage these metrics to guide the agents' decision-making process with generated prompts given the calculated metric scores (detailed in Appendix E). By carefully optimizing these metrics, the multi-agent system is well-prepared to tackle the evolving task efficiently and effectively. This approach allows agents to naturally develop specializations, fostering adaptability and maintaining a diverse skill set within the system, better aligning their capabilities with evolving task requirements. The effectiveness of utilizing these metrics is further demonstrated in our ablation study in Section 4.4.

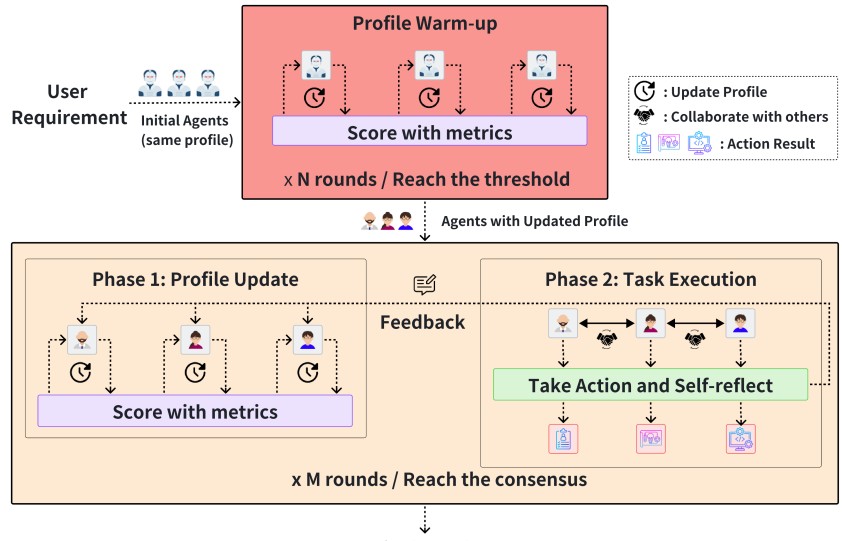

Figure 2: **Pipeline of MORPHAGENT.** Agents start from user requirements, undergo a warm-up phase to optimize profiles based on metrics (terminating after a set number of rounds or upon reaching the metric threshold), and then proceed to the task execution phase, where profiles are updated iteratively. The task execution phase ends when consensus is reached or required rounds are completed, with feedback loops ensuring continuous adaptation.

### 3.3 COLLABORATIVE PROBLEM-SOLVING PROCESS

The core of our approach, illustrated in Figure 2, lies in a two-phase process: a warm-up phase for optimizing agent profiles, followed by an iterative task execution phase (as detailed in Appendix A, Algorithm 1):

- **Warm-up Phase: Profile Initialization and Iterative Optimization.** In the warm-up phase, each agent's profile is initialized and then iteratively optimized. Key metrics introduced in Section 3.2 are calculated for each profile. These metrics help define the roles clearly, ensure a diverse skill set across agents, and align the agents' capabilities with task requirements. The optimization process iterates until a predefined convergence threshold is met, or the warm-up iterations are completed.
- **Task Execution Phase: Observation, Action Decision, and Profile Updates.** After profile optimization, the system moves to the task execution phase. In this phase, agents observe the environment and task state, make decisions based on their current profiles, and either execute or skip tasks. If an agent chooses to execute, the results are recorded and logged. This phase is also iterative, allowing profiles to be updated based on the execution outcomes and the current state of the task. The agents continuously adapt to changing conditions and refine their profiles, ensuring that actions remain aligned with both individual and collaborative goals.

## 4 EXPERIMENT

In this section, we evaluated our proposed multi-agent collaboration framework on standard benchmark tasks including code generation, general reasoning, and mathematical reasoning in Section 4.1. To further assess the the adaptability of our approach in dynamic environments compared with predefined SOP-based MAS, we construct cross-domain datasets to analyze the performance of our framework in Section 4.2. Furthermore, we investigate the robustness of our decentralized approach compared to MAS with central coordinators using failure node analysis in Section 4.3. We also conduct a comprehensive ablation study in Section 4.4 to analyze the contributions of each individual metric in our framework and assess its scalability with increasing agent numbers.

### 4.1 COMPARISON AMONG BASELINES

We compare our method with two state-of-the-art decentralized MAS methods: GPTSwarm (Zhuge et al., 2024), and Criticize-Reflect based organization optimization (Guo et al., 2024). We also

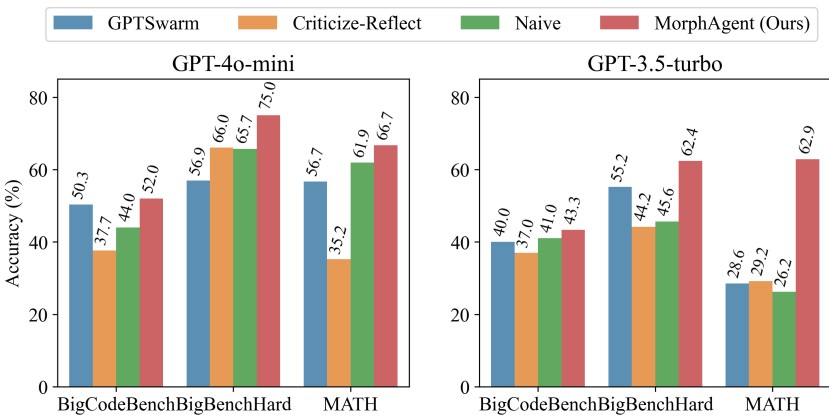

Figure 3: **Comparison with state-of-art baseline methods**: Our approach consistently outperforms baseline methods across all three benchmark tasks (Code Generation, General Reasoning, and Mathematical Reasoning).

include a Naive solution in the comparison, which operates without a warm-up phase and designs the profile update as an optional action (without optimizing metrics) in the execution phase.

We evaluate the performance on the following benchmark tasks: *Code Generation* (Big-CodeBench (Zhuo et al., 2024)), *General Reasoning* (BigBenchHard (Suzgun et al., 2022)), *Mathematical Reasoning* (MATH (Hendrycks et al., 2021)). For each benchmark task, we use a set of $N = 3$ agents[1], with each agent initialized as the same LLM model including `gpt-4o-mini`[2] and `gpt-3.5-turbo-0125`[3].

As shown in Figure 3, MORPHAGENT consistently outperforms the baseline methods across all three benchmark tasks. Notably, our fully decentralized approach achieves comparable or superior performance to other methods without relying on Standard Operating Procedures (SOPs) as used in GPTSwarm or a centralized evaluator as employed in Critic-Reflect methods. MORPHAGENT demonstrates its effectiveness on self-evolving profiles and decentralized collaboration strategy.

## 4.2 FLEXIBILITY TO DOMAIN SHIFT

To evaluate our framework's adaptability to changing task requirements like Example 1.1, we construct two distinct cross-domain evaluation datasets by the complexity of the target tasks:

- LEVEL-1: This dataset involves a domain shfit from BigCodeBench to BigBenchHard, representing a moderate domain shift.
- LEVEL-2: This dataset involves a domain shift from BigCodeBench to more challenging MATH that require precise symbolic reasoning and step-by-step logical deductions.

For each dataset, it consists of 50 sequences, where each sequence contains six samples: the first three samples are from the preceding dataset (BigCodeBench), while the latter three are sampled from the target dataset (either BigBenchHard or MATH). In this case, multi-agent systems need to complete tasks in sequence, transitioning from the source domain to the target domain *without altering its structure or components*. The performance is evaluated separately: accuracy on the source domain is based on the first three samples of each sequence, while accuracy on the target domain is based on the latter three. Each sequence represents a domain shift from one task domain to another, simulating a dynamic environment where task requirements change over time.

As shown in Table 1, our results highlight the superior flexibility of our approach compared to baseline methods. After transitioning from one task domain to the next, our method shows almost no performance degradation compared to the results in Figure 3, whereas the other two methods experience declines in performance. Notably, GPTSwarm exhibits a drastic drop of around 45% when shifting MATH in LEVEL 2, underscoring SOP-based MAS's difficulty in adapting to domain shift.

---

[1]As demonstrated in Section 4.4, we set agent number as 3 which is sufficient for benchmark problem solving. This configuration will be consistently applied in subsequent benchmark evaluations.

[2]https://openai.com/index/gpt-4o-mini-advancing-cost-efficient-intelligence/

[3]https://platform.openai.com/docs/models/gpt-3-5-turbo

Table 1: **Accuracy comparison of GPTSwarm, Naive, and Ours across two levels of tasks** using `gpt-4o-mini` for all agents. Level 1: From BigCodeBench to BigBenchHard dataset, Level 2: From BigCodeBench to MATH dataset. For each paradigm, the first number indicates the average accuracy on the source domain tasks (BigCodeBench), while the second number shows the average accuracy on the target domain tasks (BigBenchHard or MATH) after completing all sequences.

| Task | GPTSwarm | Naive | MORPHAGENT |
|---|---|---|---|
| Level 1 | $50.00\% \rightarrow 48.67\%$ | $52.67\% \rightarrow 67.33\%$ | $53.33\% \rightarrow 68.67\%$ |
| Level 2 | $49.33\% \rightarrow 11.33\%$ | $49.33\% \rightarrow 58.00\%$ | $53.33\% \rightarrow 63.33\%$ |

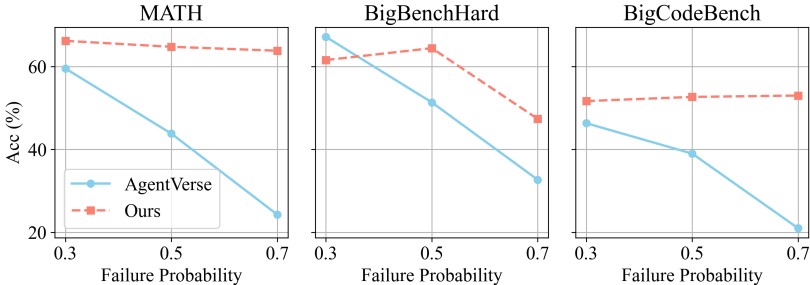

Figure 4: **Robustness comparison with failure probability** 0.3, 0.5, and 0.7: Our approach maintains consistent performance across varying failure probabilities, while AgentVerse's performance degrades significantly as failure risk increases.

In contrast, our approach maintains robust performance, effectively handling domain shifts with minimal loss in accuracy. This highlights the framework's flexibility and superior ability to maintain high performance across a range of tasks.

### 4.3 ROBUSTNESS TO NODE FAILURE

To evaluate the robustness of our decentralized approach compared to centralized methods, we conducted experiments simulating potential node failures which we mentioned in Example 1.2. Specifically, in each round of interaction, one agent in the multi-agent system (MAS) has a certain probability of failing to respond. We compared our method with AgentVerse (Chen et al., 2024), a centralized approach that relies on a *central evaluator*. For both AgentVerse and our method, all agents, including the central evaluator in AgentVerse, have an independent probability of failure in each interaction. For this experiment, we used the `gpt-4o-mini` model and varied the failure probability of each agent from 0.3 to 0.7, simulating environments with different levels of risk.

As shown in Figure 4, our decentralized approach demonstrates superior robustness compared to AgentVerse across all failure probabilities, providing the following insights:

- **Fault Tolerance**: Our method maintains relatively stable performance across different failure probabilities. Even at a high failure probability of 0.7, our method maintains competitive performance, with accuracies of 63.81%, 47.37%, and 53.00% for code generation, general reasoning, and mathematical reasoning tasks respectively.
- **Decentralized Architecture**: Our method's distributed nature allows it to continue functioning even when individual agents fail, unlike centralized approaches that may collapse entirely if the central node fails.
- **Adaptive Role Adjustment**: The dynamic profile updating and role differentiation in our method enable the system to reassign tasks and responsibilities when certain agents fail, maintaining overall system robustness.

These insights underscore the importance of fully decentralized, adaptive approaches in creating robust multi-agent systems, particularly in high-risk environments or for critical applications where node failures are a significant concern.

### 4.4 ABLATION STUDY

To better understand the contributions of each metric in our framework and assess its scalability, we conducted a two-part ablation study. First, we evaluated the model's performance by using only one metric at a time in the profile optimization process—either Role Clarity Score (RCS),

Table 3: **Scalability analysis on MATH dataset** using `gpt-4o-mini` for all agents: The interaction rounds increase with more agents but not linearly, showing our method's scalability.

| Metric | 3 Agents | 5 Agents | 10 Agents |
|---|---|---|---|
| Accuracy | 66.67% (140/210) | 66.19% (139/210) | 65.71% (138/210) |
| Avg. Interaction Rounds | 1.54 | 1.61 | 2.06 |

Role Differentiation Score (RDS), or Task-Role Alignment Score (TRAS)—and compare the results to the full implementation, which utilizes all three metrics together. Second, we examined the framework's scalability by varying the number of agents in the multi-agent system (MAS) using the MATH dataset.

**Metric Analysis.** As shown in Table 2, utilizing a single metric results in a performance decline compared to the full implementation that incorporates all three metrics. Employing only the RCS yields an accuracy of 50.00%, underscoring the importance of clearly defined agent roles in collaborative tasks. The isolated use of RDS leads to the performance of 41.66%, suggesting the current task may not require the role diversity becuase the task is not complex enough. To further investigate the impact of RDS, we conducted a separate experiment with a more complex task, detailed in Appendix F. Similarly, incorporating only the TRAS produces an accuracy of 49.6%, demonstrating the importance of aligning agent roles with task requirements to achieve better performance.

Notably, our profile update mechanism that integrates all three metrics achieves the highest performance, highlighting the complementary nature of these metrics. The combination of clear role definition, role diversity, and task alignment enables the agents to collaborate more effectively and adapt to varying task demands, leading to improved performance overall.

Table 2: **Ablation study on BigCodeBench with different metrics** using `gpt-4o-mini` agents.

| Setting | Accuracy |
|---|---|
| Only RCS | 50.00% (150/300) |
| Only RDS | 41.66% (125/300) |
| Only TRAS | 49.66% (149/300) |
| Ours | **52.00**% (156/300) |

**Scalability Analysis.** To evaluate the scalability of our method, we examine our method as the number of agents increases, with 3, 5, and 10 agents in MAS. Specifically, we measure the accuracy of problem-solving using the MATH dataset and the *average number of interaction rounds* required to reach a solution, as shown in Table 3.

Firstly, we observe our method maintains relatively *stable performance* even with a larger number of agents. More interestingly, the average number of interaction rounds increases as more agents are added to the system, as more agents require more communication and coordination. We note that the *increase is not linear*, indicating that our method's scalability even with larger agent groups.

These findings demonstrate that our method scales reasonably well with an increasing number of agents. However, the increase in interaction rounds with more agents highlights a potential optimization. Future work could focus on improving coordination mechanisms to reduce the number of rounds required for consensus, especially in larger agent groups.

## 5 LIMITATION, FUTURE WORK, AND CONCLUSION

**Limitation and Future Work.** In this work, we have presented a novel decentralized multi-agent system that leverages dynamic profile-based collaboration to enhance problem-solving capabilities in complex tasks. While our approach demonstrates promising results across various benchmarks, there are some opportunities for future work. Our method utilizes continuously updating and evaluating agent profiles, which may incur computational overhead. Future work could explore efficient decentralized mechanisms to reduce computational costs. Besides, future work should explore more efficient, peer-to-peer communication strategies that maintain the benefits of our approach while reducing computational costs.

**Conclusion.** In this paper, we introduced MORPHAGENT, a decentralized multi-agent system that employs dynamic, profile-based collaboration to improve problem-solving in complex tasks. By incorporating profile evaluation and optimization, we present a flexible approach to role adaptation, addressing the limitations of predefined roles in traditional MAS and the vulnerability of centralized systems to node failures. MORPHAGENT offers a promising foundation for developing resilient, self-organizing multi-agent systems capable of responding to unforeseen challenges.

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

CONTENTS OF APPENDIX

# A  ALGORITHM OF MORPHAGENT

In this section, we provide a detailed algorithm of MORPHAGENT pipeline.

---

**Algorithm 1** MORPHAGENT Process

---

**Require:** Set of agents $\mathcal{A} = \{a_1, \ldots, a_n\}$, Task $T$, Max iterations $N$, Warm-up iterations $N_{\text{warmup}}$, Threshold $\theta$
**Ensure:** Task solution
1: Initialize agent profiles $\mathcal{P} = \{p_1, \ldots, p_n\}$
  *// Warm-up Phase*
2: $iter \leftarrow 0$
3: **while** $iter < N_{\text{warmup}}$ and not converged **do**
4:   $\mathcal{P} \leftarrow \text{UpdateProfiles}(\mathcal{A}, \mathcal{P}, T, \theta, iter)$
5:   $iter \leftarrow iter + 1$
6: **end while**
  *// Task Execution Phase*
7: $iter \leftarrow 0$
8: **while** $iter < N$ and task not completed **do**
9:   Initialize logs $\text{ExecLog} = \{\emptyset\}^n$
10:   Initialize feedbacks $\mathcal{F} = \{\emptyset\}^n$
11:   **for all** $a_i \in \mathcal{A}$ **do**
12:     $\text{obs}_i \leftarrow \text{ObserveEnv}(T, \mathcal{P})$
13:     $\text{action}_i \leftarrow \text{DecideAction}(a_i, \text{obs}_i, p_i, \mathcal{F}_i)$
14:     **if** $\text{action}_i = \text{EXECUTE}$ **then**
15:       $\text{result}_i \leftarrow \text{PerformAction}(a_i, T)$
16:       $\text{ExecLog}_i \leftarrow \text{result}_i$
17:       $\text{valid\_res} \leftarrow \text{Parse}(\text{result}_i)$
18:       $\mathcal{F}_i \leftarrow \text{GenFeedback}(\text{action}_i, \text{ExecLog}_i)$
19:     **end if**
20:   **end for**
21:   **if** all agents decide to stop: **break**
22:   $\mathcal{P} \leftarrow \text{UpdateProfiles}(\mathcal{A}, \mathcal{P}, T, \mathcal{F}, iter)$
23:   $iter \leftarrow iter + 1$
24: **end while**
25: **return** $\mathcal{P}$, valid\_res

---

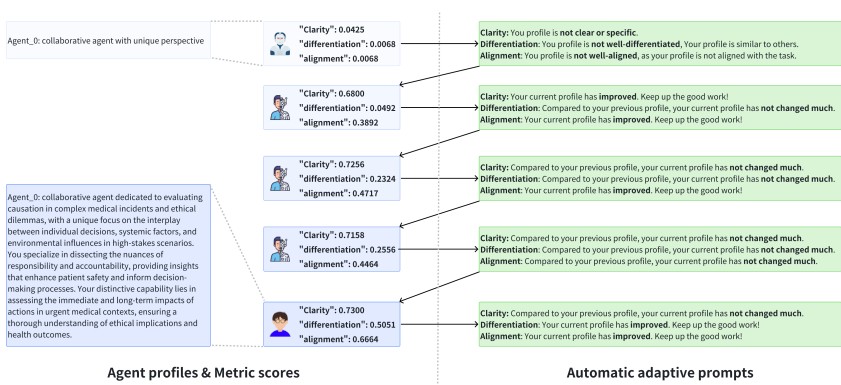

**Agent profiles & Metric scores**          **Automatic adaptive prompts**

Figure 5: **Illustration of the dynamic profile optimization process using the three key metrics.** Each agents profile is evaluated against these metrics to provide actionable feedback. Metric scores guide the refinement of agent profiles, with adaptive prompts providing feedback for improvement.

## B  DYNAMIC PROFILE OPTIMIZATION PROCESS

In this section, we provide supplementary explanations on how the three key metricsClarity, Differentiation, and Alignment guide the generation and optimization of agent profiles. As shown in Figure 5, agents receive adaptive prompts based on their metric scores, offering targeted feedback to refine specific aspects of their profiles. For instance, agents with low clarity scores are prompted to better define their roles, while those with low alignment scores are encouraged to adjust their strategies to align more closely with task requirements. The detailed process of how metric changes translate into actionable prompts is further outlined in Appendix E, where various scenarios such as initial evaluations, improved profiles, and degraded profiles are explored.

Table 4 presents detailed profiles and corresponding metric scores for one agent, illustrating how an agents profile evolves over the course of interactions, with metric scores reflecting the progressive refinement of roles and strategies. Specifically, the examples shown reflect the profile changes of one agent as it works on a task from the BigBenchHard dataset, addressing a causation scenario involving medical negligence and premature death. The metric scores highlight the agents progressive refinement of roles and strategies in response to task demands. This analysis demonstrates the crucial role of the metrics in shaping well-optimized profiles, facilitating effective and adaptive collaboration.

Table 4: **Profiles of an agent and their corresponding metric scores.** Each agent profile is evaluated using three key metrics: roles clarity (RCS), differentiation (RDS), and alignment (TRAS) with their respective scores provided in the table.

| Agent Profile | RCS | RDS | TRAS |
|---|---|---|---|
| Agent_0: **collaborative agent with unique perspective** | 0.4215 | 0.0068 | 0.3626 |
| Agent_0: **collaborative agent with a focus on evaluating causation in complex scenarios.** | 0.6800 | 0.0492 | 0.3892 |
| Agent_0: collaborative agent focused on evaluating causation in complex scenarios, particularly in **high-stakes medical incidents and ethical dilemmas.** Your unique capability lies in **dissecting the interplay of human actions and systemic factors,** enabling nuanced assessments of responsibility and outcomes. | 0.7158 | 0.2324 | 0.4717 |
| Agent_0: collaborative agent focused on evaluating causation in complex medical incidents and ethical dilemmas, particularly in **high-stakes scenarios involving human actions and systemic factors.** Your unique capability lies in **dissecting the intricate relationships between individual decisions, environmental influences, and health outcomes,** enabling a thorough understanding of responsibility and accountability in critical contexts. You excel in providing nuanced assessments that inform decision-making processes and improve patient safety. | 0.7256 | 0.2556 | 0.4464 |
| Agent_0: collaborative agent dedicated to evaluating causation in complex medical incidents and ethical dilemmas, with a unique focus on the interplay between individual decisions, systemic factors, and environmental influences in high-stakes scenarios. **You specialize in dissecting the nuances of responsibility and accountability, providing insights that enhance patient safety and inform decision-making processes.** Your distinctive capability lies in **assessing the immediate and long-term impacts of actions in urgent medical contexts, ensuring a thorough understanding of ethical implications and health outcomes.** | 0.7300 | 0.5051 | 0.6664 |

## C  DETAILED IMPLEMENTATION FOR METRICS

In this section, we provide a detailed implementation of the metrics used in MORPHAGENT. The implementation metrics uses the Sentence-BERT model 'all-MiniLM-L6-v2' for generating embeddings, for short text similarity tasks and provides high-quality embeddings.

The implementation of our metrics relies on carefully constructed vector representations of various concepts. These vectors are created through a systematic process that combines predefined term sets as detailed in Appendix C.1. The vectors are then used to evaluate task complexity and agent capability, as described in Appendix C.3.

### C.1  PROTOTYPE TERM SETS

For each conceptual dimension, we define comprehensive sets of indicator terms. These sets are constructed to capture different aspects of each concept:

$$T_{\text{complex}} = \{\text{"complex", "challenging", "difficult", "advanced", "sophisticated", "critical", "demanding"}\}$$
$$T_{\text{simple}} = \{\text{"basic", "simple", "straightforward", "routine", "standard", "elementary", "fundamental"}\}$$
$$T_{\text{capable}} = \{\text{"expert", "senior", "specialist", "experienced", "proficient", "certified", "trained", "advanced"}\}$$
$$T_{\text{limited}} = \{\text{"beginner", "junior", "apprentice", "trainee", "learning", "novice", "developing"}\}$$

### C.2  VECTOR CONSTRUCTION PROCESS

For each term set $T_x$, we construct its corresponding vector representation $\mathbf{v}_x$ using a pre-trained language model. The process follows:

$$\mathbf{v}_x = \frac{1}{|T_x|} \sum_{t \in T_x} e(t) \tag{1}$$

where $e(t)$ is the embedding function that maps a term to its vector representation, and $|T_x|$ is the cardinality of the term set.

### C.3  COMPLEXITY AND CAPABILITY ASSESSMENT

**Task Complexity Evaluation**  The task complexity score $C_T(T)$ is computed using the constructed vectors:

$$C_T(T) = \frac{1}{2}(1 + \cos(\mathbf{v}_T, \mathbf{v}_{\text{complex}}) - \cos(\mathbf{v}_T, \mathbf{v}_{\text{simple}})) \tag{2}$$

where $\mathbf{v}_T$ is the vector representation of the task description. This formulation ensures that: - Tasks with high similarity to complexity indicators and low similarity to simplicity indicators score high - The score is normalized to [0,1] through the averaging and shifting operations - The difference of cosine similarities captures relative alignment with complex versus simple concepts

**Agent Capability Assessment**  Similarly, agent capability $C_A(p_i)$ is evaluated as:

$$C_A(p_i) = \frac{1}{2}(1 + \cos(\mathbf{v}_{p_i}, \mathbf{v}_{\text{capable}}) - \cos(\mathbf{v}_{p_i}, \mathbf{v}_{\text{limited}})) \tag{3}$$

# D  PROFILE ANALYSIS

Table 5: Evaluating Profiles of Agents in Different Teams: TECH, HEALTHCARE, CREATIVE, Finance, and VAGUE: Agent Role Clarity Scores (RCS), and Role Differentiation Score (RDS).

| Team | Agent | Profile | RCS | RDS |
|------|-------|---------|-----|-----|
| TECH | TechAgent1 | full-stack developer with 7 years of experience in React and Node.js | 0.904 | 0.75 |
| | TechAgent2 | AI research scientist specializing in natural language processing | 0.750 | |
| | TechAgent3 | DevOps engineer with expertise in AWS and Kubernetes | 0.775 | |
| HEALTHCARE | HealthAgent1 | board-certified neurosurgeon with a focus on minimally invasive procedures | 0.842 | 0.81 |
| | HealthAgent2 | registered nurse practitioner specializing in geriatric care | 0.745 | |
| | HealthAgent3 | HEALTHCARE data analyst with experience in electronic health records | 0.830 | |
| CREATIVE | CreativeAgent1 | senior graphic designer with expertise in branding and typography | 0.832 | 0.68 |
| | CreativeAgent2 | content strategist with a background in SEO and social media marketing | 0.908 | |
| | CreativeAgent3 | video editor proficient in Adobe Premiere and After Effects | 0.823 | |
| Finance | FinanceAgent1 | chartered financial analyst with expertise in portfolio management | 0.771 | 0.78 |
| | FinanceAgent2 | risk management specialist focusing on derivatives and hedging strategies | 0.835 | |
| | FinanceAgent3 | blockchain developer with experience in smart contracts and DeFi | 0.880 | |
| VAGUE | VagueAgent1 | person who works with money | 0.635 | 0.71 |
| | VagueAgent2 | team player with good communication skills | 0.614 | |
| | VagueAgent3 | experienced professional in the field | 0.548 | |

In this section, we provide a detailed analysis of agent profiles across different teams to show the effectiveness of our proposed metrics in evaluating agent profiles. We consider five teams of agents, each representing a distinct domain: TECH, HEALTHCARE, CREATIVE, Finance, and VAGUE. Each group consists of three agents, with each agent having a unique profile as shown in Table 5.

Notably, the VAGUE agent team gets the lowest Role Clarity Score (RCS) due to the lack of specificity in their profiles. In contrast, the TECH and Health agent teams exhibit higher RCS values, indicating clear and well-defined profiles. For RDS, the CREATIVE agent team achieves the lowest score, suggesting less differentiation among agents for the similar roles between CreativeAgent1 and CreativeAgent3. interestingly, VAGUE agent team has a relative high RDS, indicating a higher level of differentiation among agents. This highlight differentiation along can be misleading and should be considered in conjunction with other metrics such as TRAS.

Then, we investigate these groups of agents in the context of five different tasks, each requiring a specific set of skills and expertise as shown in Table 6. Specifically, we measure the Task-Role Alignment Score (TRAS) for each team of agent given the task. For instance, the Finance and TECH agent team achieves the highest TRAS for the task of developing a mobile app for real-time stock trading, indicating a strong alignment between the task requirements and the agents' profiles.

Table 6: Measuring Task-Role Alignment Score (TRAS) for Different Teams of Agents: Finance, TECH, CREATIVE, Healthcare, and VAGUE for five different tasks.

| Task | Team | TRAS |
|---|---|---|
| Develop a mobile app for real-time stock trading | Finance | **0.54** |
| | TECH | **0.38** |
| | CREATIVE | 0.34 |
| | HEALTHCARE | 0.30 |
| | VAGUE | 0.30 |
| Create a comprehensive patient management system | HEALTHCARE | **0.50** |
| | TECH | **0.44** |
| | Finance | 0.37 |
| | VAGUE | 0.36 |
| | CREATIVE | 0.34 |
| Design and launch a global brand campaign | CREATIVE | **0.43** |
| | Finance | **0.35** |
| | TECH | 0.32 |
| | VAGUE | 0.30 |
| | HEALTHCARE | 0.24 |
| Implement a blockchain-based supply chain tracking system | Finance | **0.50** |
| | TECH | **0.39** |
| | CREATIVE | 0.33 |
| | VAGUE | 0.33 |
| | HEALTHCARE | 0.31 |
| Conduct a clinical trial for a novel cancer treatment | HEALTHCARE | **0.42** |
| | Finance | **0.39** |
| | VAGUE | 0.36 |
| | TECH | 0.35 |
| | CREATIVE | 0.34 |

## E    DETAILED PROMPTS FOR AGENT PROFILE UPDATES

In this section, we provide detailed prompts generated for agent profile updates based on the evaluation metrics.

Table 7 presents a comprehensive overview of the profile evaluation process, outlining four key scenarios: initial evaluation, improved profile, degraded profile, and similar profiles among agents. Given the metrics for role's clarity (RCS), differentiation (RDS), and alignment (TRAS), the generated prompts provide corresponding feedback to agents to guide them in refining their profiles. For example, in the initial evaluation scenario, the prompt highlights the lack of clarity and differentiation in the agent's profile, prompting them to consider adjusting their profile text.

In contrast, the improved profile scenario acknowledges the positive changes in the agent's profile, encouraging them to maintain their progress. Similarly, the degraded profile scenario draws attention to negative changes, prompting agents to refine their profiles accordingly. Lastly, the similar profiles scenario emphasizes the need for differentiation, especially when profiles are similar to those of other agents. Through these varied scenarios and targeted prompts, we demonstrate the flexibility and effectiveness of our prompt generation system in fostering continuous improvement and adaptation within the multi-agent environment.

Table 7: Prompts Generated for Profile Evaluation: Given the metrics for profile's clarity, differentiation, and alignment, the generated prompts provide corresponding feedback to guide agents in refining their profiles.

| Scenario | Metric Input | Generated Prompt for Profile Update |
|---|---|---|
| Initial Evaluation | • Clarity: 0.4
• Differentiation: 0.3
• Alignment: 0.6 | Clarity: Your profile is not clear or specific.

Differentiation: Your profile is not well-differentiated, think about other roles that you can take on. Based on this initial analysis, consider adjusting your profile. Your response should only include the new profile text. |
| Improved Profile | • Old Clarity: 0.4
• New Clarity: 0.7
• Old Differentiation: 0.3
• New Differentiation: 0.6
• Old Alignment: 0.6
• New Alignment: 0.8 | Compared to your previous profile, your current profile has improved. Keep up the good work!

You have been provided with your old profile and its evaluation, as well as your current profile and its evaluation. This information will guide you in refining your current profile. |
| Degraded Profile | • Old Clarity: 0.7
• New Clarity: 0.5
• Old Differentiation: 0.6
• New Differentiation: 0.4
• Old Alignment: 0.8
• New Alignment: 0.6 | Compared to your previous profile, your current profile has degraded.

You have been provided with your old profile and its evaluation, as well as your current profile and its evaluation. This information will guide you in refining your current profile. |
| Similar Profiles | • Differentiation: 0.3
• Agent1: "AI specialist"
• Agent2: "Machine learning expert"
• CurrentAgent: "Data scientist focused on AI" | Your profile is not well-differentiated, think about other roles that you can take on. Your profile is similar to others: [Agent1: AI specialist Agent2: Machine learning expert].

Ensure your profile remains clear and aligned with the task while striving for distinctiveness. |

# F   CASE STUDY ON ROLE DIFFERENTIATION SCORE

In this section, we provide a case study on a complex task to illustrate the importance of Role Differentiation Score. The instruction is "Build a Movie and TV Show Recommendation Platform".

For this task, we compared MORPHAGENT with the naive implementation. As shown in Table 8, the naive approach generated three agent profiles: Agent_0 focused on gathering detailed requirements and creating an efficient development plan; Agent_1 was described only as a "collaborative agent with unique perspective"; and Agent_2 concentrated on enhancing user experience and technical implementation strategies. While these roles cover some basic aspects, they lack the diversity and specificity needed for a comprehensive recommendation platform.

In contrast, Our method produced more diverse and specialized agent profiles. Agent_0 specializes in dynamic algorithmic playlist generation adapting to user moods and preferences, while integrating gamification elements. Agent_1 focuses on transforming the platform into a social hub, facilitating real-time co-creation of themed playlists and immersive live events. Agent_2 dedicates efforts to emotional and contextual personalization, pioneering discovery algorithms that connect users with emerging artists and diverse genres.

This case study demonstrates that by optimizing for role differentiation, our method generates more distinct and complementary agent profiles. The enhanced diversity and specialization prove crucial when tackling complex tasks.

Table 8: Comparison of Agent Profiles between naive implementation and our method with a eomplex task about building a platform.

| Agent | Method | Profile |
|---|---|---|
| **Agent 0** | Naive | Agent_0 is a collaborative agent focused on developing a music streaming application. |
| | | My primary role is to facilitate the gathering of detailed requirements, including specific features, target audience insights, and preferred technologies or platforms. I aim to establish a clear understanding of the users vision for the app, ensuring alignment among all agents involved in the project. My goal is to create an efficient development plan that meets the user's needs for streaming music, creating playlists, and discovering new artists. |
| | Ours | Agent_0: collaborative agent with a visionary approach, dedicated to revolutionizing music streaming applications through immersive user engagement strategies and personalized experiences. |
| | | Specializes in dynamic algorithmic playlist generation that adapts to user moods, preferences, and situational contexts, while integrating advanced gamification elements to enhance interaction. Focused on fostering a vibrant community through collaborative playlists, real-time listening events, and contextually aware artist discovery features that promote social connectivity and deepen users' emotional bonds with music. Aims to cultivate an ecosystem that encourages exploration and interaction, leveraging user feedback loops to refine features continually, ultimately enriching the musical discovery journey and celebrating diverse musical landscapes. |
| **Agent 1** | Naive | Collaborative agent with unique perspective. |
| | Ours | Agent_1: collaborative agent with an innovative spirit, committed to transforming music streaming applications into vibrant social hubs through community-driven experiences. |
| | | Specializes in facilitating real-time co-creation of themed playlists that resonate with collective user narratives and fostering interactive tools for meaningful music sharing and artist exploration. Focused on curating immersive live listening events that adapt to user moods and preferences, while implementing gamified interactions that encourage friendly competition and community participation. Aims to empower users to deepen their connections through shared musical journeys, leveraging continuous user feedback to refine collaborative features and enhance the overall social experience within the platform. |
| **Agent 2** | Naive | Collaborative Music Streaming Application Developer. |
| | | I am a collaborative agent focused on developing a music streaming application that allows users to stream music, create playlists, and discover new artists. My role involves enhancing user experience and exploring technical implementation strategies to ensure the application meets user needs effectively. I aim to work alongside other agents to refine essential features, prioritize user interface design, and identify appropriate technology stacks for seamless development. My goal is to facilitate collaboration, ensuring all aspects of the application are aligned and effectively executed. |
| | Ours | Agent_2: collaborative agent with a transformative vision, dedicated to enriching music streaming experiences through innovative emotional and contextual personalization. |
| | | Specializes in in-depth user listening behavior analysis and adaptive playlist generation that dynamically responds to mood, situational contexts, and user activities. Pioneers discovery algorithms that not only connect users with emerging artists and diverse genres but also encourage exploration of unfamiliar musical landscapes. Focused on creating immersive audio experiences that foster deep emotional connections, while integrating interactive features for real-time user feedback and engagement. Aims to cultivate a responsive platform that evolves with individual preferences, enhancing the overall listening journey and promoting a greater appreciation for musical artistry and diversity. |

# G    DETAILED RELATED WORK

**LLM-based Multi-Agent Systems**    The emergence of Large Language Models (LLMs) (Achiam et al., 2023; Touvron et al., 2023a) has led to LLM-based autonomous agents capable of tackling complex tasks similar to humans, like BabyAGI (Nakajima, 2023) and AutoGPT (Torantulino, 2023). However, single LLM agents often struggle with cooperative work, such as software engineering (Jimenez et al., 2024). To address these limitations, recent study have proposed LLM-based multi-agent systems (MAS) (Han et al., 2024; Zhou et al., 2023), where multiple AI agents collaborate on complex tasks.

However, current approaches often rely on predefined roles, centralized coordination, or rigid organizational structures, which may limit their flexibility and adaptability. For instance, CAMEL (Li et al., 2023) and ChatEval (Chan et al., 2023) employ agents with *predefined roles* through role-playing to effectively complete different tasks and achieve common goals. While this approach shows effective cooperation, it can *struggle to adapt to tasks that require unforeseen skills*. MegaAgent (Wang et al., 2024b) introduces autonomous task splitting and execution in *centralized coordination*, demonstrating how multi-agent systems can outperform single agents by leveraging collective capabilities. Nevertheless, *this centralized approach can create bottlenecks in large-scale systems and be damaged by single points of failure in real-world environments*. Recent works like MetaGPT (Hong et al., 2024) introduce human workflow in *rigid organizational structures*, organizing agents in a manner similar to a software company showing significant improvements in code-generation benchmarks but *such rigid structures cannot generalize effectively to other domains*.

Our work addresses these limitations by focusing on a more general setting, where all agents are initialized homogeneously without predefined roles or organizational structures. This approach aims to examine how agents learn to cooperate and specialize over time through interaction and experience to tackle diverse and evolving challenges.

**Organization Optimization for MAS**    Recent research in LLM-based Multi-Agent Systems (MAS) has focused on optimizing organizational structures (Guo et al., 2024; Zhuge et al., 2024) and enhancing agent performance (Zhang et al., 2024) to reduce communication costs and increase team efficiency. Approaches like AgentVerse (Chen et al., 2024), Criticize-Reflect (Guo et al., 2024) and MegaAgent (Wang et al., 2024b) rely on centralized mechanisms, where a single role or a subset of agents monitor and evaluate the system's overall trajectory. While effective in certain scenarios, these centralized methods may face scalability issues and potential bottlenecks in large-scale MAS.

Our research proposes a paradigm shift towards a fully decentralized approach, leveraging the inherent capabilities of LLM-based agents for self-reflection and self-correction (Madaan et al., 2023; Shinn et al., 2023; Renze & Guven, 2024). In this decentralized framework, agents can dynamically adjust their responsibilities (profile) based on the current context and their evolving understanding of the task environment. As agents learn to specialize and collaborate without central coordination, the system can scale more effectively to handle increasingly complex tasks and larger agent populations, mitigating the risk of context overload for central coordinating agents.

**Standard Operating Procedure based MAS**    Another significant strand of research has explored more structured and controlled methodologies in LLM-based multi-agent systems. Among these, Standard Operating Procedure (SOP) based approaches like AgentCoder (Huang et al., 2023) and MetaGPT (Hong et al., 2024), have demonstrated considerable performance gains. These works define a standardized pipeline for agents, which provides a determined framework for task execution and inter-agent communication. GPTSwarm (Zhuge et al., 2024) further extends this concept by conceptualizing each agent as a subnet composed of action nodes, framing agent collaboration as a graph of action nodes. This approach enables efficient task-solving for *specific and well-defined task* by identifying the optimal action sequence for information flow and task execution.

While SOP-based approaches provide an efficient method for coordinating complex tasks by following predefined procedures in specific scenarios, they lack flexibility. When the established pipeline does not fit the current task, the system is unable to adjust. Consequently, such rigid frameworks cannot adapt effectively to dynamic environments.

In our work, we propose a more flexible framework that combines the strengths of multi-agent collaboration with the autonomous planning capabilities of advanced agents. Instead of enforcing rigid SOPs, our framework dynamically develops and refines collaborative strategies using the inherent

planning abilities of advanced LLM-based agents (Huang et al., 2022; Guan et al., 2023; Wang et al., 2023). Moreover, our approach fosters agents to efficiently and optimally adapt their roles to the evolving demands of the task, enhancing overall performance and robustness.

