# OpenReview forum: "MorphAgent: Empowering Agents through Self-Evolving Profiles and Decentralized Collaboration"
_ICLR.cc/2025/Conference — Submitted to ICLR 2025_

### Official Review · Reviewer_uHtB · 2024-10-21

**Soundness:** 3
**Presentation:** 3
**Contribution:** 3
**Rating:** 6
**Confidence:** 4

**Summary:**

MORPHAGENT is a fully decentralized multi-agent system that enables agents to autonomously adapt their roles and capabilities through self-evolving profiles optimized using three key metrics: Role Clarity Score, Role Differentiation Score, and Task-Role Alignment Score. The framework employs a two-phase process—a warm-up phase for initial profile optimization and a task execution phase where agents iteratively update their profiles based on task feedback—enhancing the system's adaptability and robustness in dynamic environments without relying on predefined roles or centralized coordination. Experimental results demonstrate that MORPHAGENT outperforms traditional static-role multi-agent systems in task performance and adaptability, effectively handling domain shifts and node failures.

**Strengths:**

I will preface this review by saying that this is not my area of expertise, therefore I might be unfamiliar with crucial work in the state of the art, making it difficult for me to fairly asses the contribution.

1. **Experimental results**: The experimental results are strong and demonstrate the advantages of using MORPHAGENT for tasks that require coordination especially when centralization might lead to issues (due to failure of important nodes) or there is domain switch.

2. **Motivation**: Decentralized systems are particularly useful in real-world scenarios where failure of specific nodes might cause the entire system to fail, therefore MORPHAGENT stands out as a promising approach for complex environments.

3. **Novelty**: The paper addresses an under-explored problem and proposes a very unique solution that is demonstrated to work in the evaluation scenarios.

**Weaknesses:**

1. **Computational overhead**: My main issue with this paper is that even though the computational overheads are aknowledged in the limitations section, they are not directly stated. In particular how much more computation is being used in wall-clock time v.s. the baselines? Without it, it is difficult to asses how applicable and practical the method really is.

2. **Clarity**: The writing of the paper is not super clear, it took me a long time to understand some of the metrics because fundamental definitions and terms are missing. In particular in dependency score, the definition of "subtree" is missing and since there are no references to Dependency Parsing, it was hard to infer that subtree referred to the dependency subtree. Similarly terms like "skill prototype" and "potential skill tokens" are used for metric definitions but not defined. More importantly, there is no intution on why the metrics are chosen, making some of them seem arbitrary in the context of role ambiguity (e.g. why is the dependency score correlated to the specificity of the profile).

3. **Fairness of the baseline comparisson**: This is a relatively minor issue, but GPTSwarm is evaluated in the GAIA Benchmark, so why not use GAIA here as well? The lack of this comparisson makes it difficult for me to assess wether the strength of MORPHAGENT is dependent on dataset specifics.

**Questions:**

1. How does MORPHAGENT handle communication between agents?
2. How did you determine the weighting coefficients $(\beta_1, \beta_2, \beta_3)$ in the Role Clarity Score? Are these weights task-specific, or did you find a set of weights that work well across different tasks?

---

> ### Author Response · Authors · 2024-11-22
> **Response to Reviewer uHtB (1/3)**
>
> > W1: Computational overhead: My main issue with this paper is that even though the computational overheads are aknowledged in the limitations section, they are not directly stated. In particular how much more computation is being used in wall-clock time v.s. the baselines? Without it, it is difficult to asses how applicable and practical the method really is.
> >
>
> Thank you for raising this important point about computational costs. While we don't have wall-clock time comparisons, we have tracked the API costs when using GPT-4-mini on MATH dataset for different methods:
>
> | GPTSwarm | 56.70% | $0.27 |
> | --- | --- | --- |
> | Criticize-Reflect | 35.24% | $6.31 |
> | Naive | 61.90% | $0.62 |
> | Ours | 66.67% | $1.02 |
>
> While our cost is higher than GPTSwarm's, it's important to note that GPTSwarm's results use their pre-optimized collaboration structures specifically designed for these tasks, bypassing the cost of discovering effective collaboration patterns. Despite this, our method achieves better performance with fourth the cost.
>
> Compared to Criticize-Reflect, another self-coordinating MAS, we achieve both better performance and significantly lower cost (about 1/6th). This demonstrates our method's efficiency in balancing performance and computational overhead.

---

> ### Author Response · Authors · 2024-11-22
> **Response to Reviewer uHtB (2/3)**
>
> > W2: Clarity: The writing of the paper is not super clear ...
>
> We apologize for any lack of clarity in our metric definitions. We have enhanced the explanations in our revised paper (highlighted for easy reference). Let me clarify some details:
>
> 1. The definitions related to Dependency Parsing
>     1. **Semantic Relationships**: Dependency relations, particularly for subjects (nsubj), objects (dobj), and prepositional objects (pobj), often encode key role responsibilities and requirements. For example: "develops (head) → software systems (dobj)" captures a core responsibility
>     2. **Syntactic Complexity**: Dependency trees capture the **hierarchical relationships between words**, reflecting the structural complexity of role descriptions. This builds on established work in syntactic parsing [1] and role analysis [2]. More detailed and specific role definitions typically exhibit:
>         - Deeper syntactic structures
>         - More complex modifier relationships
>         - Richer argument structures
>     3. For example,
>         - Role Description 1 (Basic): "Develops software applications."
>
>             Key subtrees [Total dep_score ≈ 0.33 (normalized)]:
>
>             - develops: size 3 (develops, software, applications)
>             - applications: size 1 (applications)
>         - Role Description 2 (Detailed): "A senior engineer develops scalable cloud-based software applications and implements robust security protocols for distributed systems."
>
>             Key subtrees [Total dep_score ≈ 0.85 (normalized)]:
>
>             - develops: size 6 (develops, engineer, senior, applications, software, cloud)
>             - implements: size 5 (implements, protocols, security, robust, systems, distributed)
>             - applications: size 2 (applications, cloud)
>             - protocols: size 2 (protocols, security)
>             - systems: size 2 (systems, distributed)
>
>         The **higher** dep_score for the second description quantitatively **reflects its greater specificity and clarity**, demonstrating how dependency analysis effectively captures role detail levels.
>
> 2. The explanation of "skill prototype" and "potential skill tokens”
>     1. Skill Prototype $s$:
>         - A vector representation capturing skill-related concepts
>         - Computed as the average embedding of skill-indicator terms (e.g., "skill", "expertise", "proficiency", "competence")
>         - Formula: $s = \frac{1}{n}\sum_{i=1}^n e(w_i)$
>     2. Potential Skill Tokens $\mathcal{PS}(p)$: These are identified through both semantic and syntactic criteria:
>         - $\mathcal{PS}(p)$ represents tokens in profile $p$ that **likely** **describe specific skills**. These are identified through both **syntactic and semantic criteria**:
>             - Semantic criteria: tokens with high similarity to the skill prototype vector
>             - Syntactic criteria: tokens that are either:
>                 - Proper nouns (PROPN) or common nouns (NOUN)
>                 - In specific dependency relations (compound, dobj, pobj)
>
>             This definition allows us to capture both **explicit skill** mentions (e.g., "Python programming") and **implicit skill** indicators (e.g., "system architecture design").
>
>         - To illustrate with an example: Given profile text: "Expert in Python programming with system architecture design experience", potential skill tokens would include: ["Python", "programming", "system architecture", "design"]
> 3. The intuition of three metrics
>     1. **Dependency Score & Role Specificity**:
>         - More specific roles naturally form deeper dependency structures
>         - Example:
>             - Vague: "Handles data tasks" (shallow structure)
>             - Specific: "develops scalable enterprise solutions for financial systems" (deep dependencies between terms reflect detailed responsibility definition)
>         - **This mirrors how human experts evaluate role descriptions: more specific roles require more structured relationships between components.**
>     2. **Role Clarity Components**
>
>         Our three-part metric (dependency, entropy, skill) maps to established dimensions of role clarity from:
>
>         - Task clarity (dependency structure)
>         - Scope definition (lexical diversity/entropy)
>         - Required capabilities (skill identification)
>
>     3. **Task-Role Alignment**
>
>     The metric is measured by:
>
>     - Semantic alignment (**what needs to be done**)
>     - Capability matching (**can it be done**)
>
> [1] Kübler, S., McDonald, R., & Nivre, J. (2009). "Dependency Parsing"
>
> [2] Jurafsky, D., & Martin, J. H. (2023). "Speech and Language Processing"

---

> ### Author Response · Authors · 2024-11-22
> **Response to Reviewer uHtB (3/3)**
>
> > W3: Fairness of the baseline comparisson: This is a relatively minor issue, but GPTSwarm is evaluated in the GAIA Benchmark, so why not use GAIA here as well? The lack of this comparisson makes it difficult for me to assess wether the strength of MORPHAGENT is dependent on dataset specifics.
> >
>
> While we acknowledge your interest in GAIA benchmark results, our current dataset selection (BigCodeBench, MATH, BigBenchHard) comprehensively covers three fundamental task types in LLM-based MAS applications: **coding, mathematical reasoning, and general reasoning**. These widely-used benchmarks provide a robust evaluation of our method's capabilities.
>
> While GPTSwarm used GAIA, it remains a relatively niche dataset that is rarely adopted in major multi-agent system evaluations. Most prominent baselines in the field (like AentVerse) have primarily used benchmarks similar to our current selection for comprehensive evaluation.
>
> > Q1: How does MORPHAGENT handle communication between agents?
> >
>
> In our implementation,
>
> - Agent communication follows a **broadcast model** where messages from any agent are visible to all other agents, allowing for flexible response patterns.
> - All agent actions and environmental feedback are stored in memory for future reference and decision-making.
> - This approach enables open communication while maintaining a structured record of interactions.
>
> > Q2: How did you determine the weighting coefficients  $(\beta_1, \beta_2, \beta_3)$ in the Role Clarity Score? Are these weights task-specific, or did you find a set of weights that work well across different tasks?
> >
>
> In our current implementation, we use equal weights (1/3 for each component) in the Role Clarity Score as a baseline approach.
>
> While determining optimal weights is not the core contribution of our work, the RCS framework is designed to support adaptive weighting.  The weights can be tuned through: domain expert input, empirical validation on specific task sets, and role-specific optimization, etc.
>
> The weight optimization can be one direction for future work, where domain-specific studies could determine optimal weight configurations for different contexts.

---

> > ### Comment · Reviewer_uHtB · 2024-11-23
> >
> > I would like to thank the authors for answering my concerns and questions. I believe that the modifications made to the manuscript significantly enhance the clarity of this work. I don't think that the contribution of this work grants me increasing my score, but the discussion has definitely increased my confidence in this paper.

---

### Official Review · Reviewer_9vnk · 2024-11-01

**Soundness:** 2
**Presentation:** 1
**Contribution:** 2
**Rating:** 5
**Confidence:** 4

**Summary:**

This paper introduces MorphAgent, a framework for decentralized multi-agent LLM collaboration that enables agents to dynamically evolve their roles and capabilities. Unlike existing approaches that rely on predefined roles or centralized coordination, MORPHAGENT employs self-evolving agent profiles optimized through three metrics. The framework implements a two-phase process: a warm-up phase for initial profile optimization, followed by a task execution phase where agents continuously adapt their roles based on task feedback. Through evaluations on various benchmarks, the authors demonstrate that MorphAgent outperforms traditional static-role systems and shows better adaptability to domain shifts and robustness to node failures.

**Strengths:**

1. The paper effectively communicates its ideas through clear visualization - Figure 1 illustrates the key challenges with concrete examples, while Figure 2 provides a comprehensive overview of the framework's workflow.
2. The experimental results seem good, showing MorphAgent's consistent performance gain across different benchmarks.
3. Analyses of their framework's advantages is presented.

**Weaknesses:**

1. The implementation details and methodology are severely unclear and poorly explained:
   - The profile updating process is vaguely described, with crucial details buried in figures and appendix
   - The three metrics are defined with numerous undefined notations and unexplained components (e.g., *skill prototype* and *potential skill tokens* in Definition 3.1, and *vector representations* in Definition 3.3)
   - The design choices lack justification, such as using dependency trees in RCS
   - The auxiliary agent is only mentioned in Section 3.1. Why is it necessary? What's the disadvantage of letting autonomous agent directly interact with the environment?
   - Experimental settings in Sections 4.2 and 4.3 are incomprehensible - the domain shift setup and node failure mechanism are not properly explained. I can't even know how these two experiments are conducted.
   - There are too many things that are not clearly explained. I've tried to list them, but there is definitely something else missing for a reader to fully understand the framework.

2. The experimental results presentation has some issues:
   - Table 1 is poorly presented with unexplained notations. I don't know what are the two numbers represent in each cell.
   - The reported improvement on MATH dataset with MorphAgent (over 30 points!) with GPT-3.5-turbo is suspiciously large and lacks explanation. It is nearly impossible for me that multi-agent debate can lead to such a significant improvement.
   - The explanation of the level in the caption of Table 1 is inconsistent with the text content.
   - The analysis of results is superficial, lacking a detailed discussion of why the method works

3. The paper lacks concrete examples and case studies:
   - No examples showing how agent profiles evolve through iterations
   - No comparison of actual responses between MorphAgent and baselines

4. The evaluation methodology is questionable:
   - The node failure experiments lack clear description of failure mechanisms. How did you incur the node failure? What does node failure mean?
   - Domain shift experiments don't clearly specify whether it's transfer learning or continuous adaptation. Is it that a multi-agent team obtained through optimization on one task is transferred to another task?

Overall, while the paper presents an interesting idea, the poor explanation of implementation details, questionable result presentation, and lack of concrete examples make it difficult to assess the true value and reproducibility of the work.

**Questions:**

See weaknesses above.

---

> ### Author Response · Authors · 2024-11-22
> **Response to Reviewer 9vnk (1/5)**
>
> > W1: The implementation details and methodology are severely unclear and poorly explained:
> >
>
> We have enhanced the clarity of the explanations in our revised manuscript (highlighted for easy reference). And Here are some detailed explanations:
>
> > The profile updating process is vaguely described, with crucial details buried in figures and appendix
> >
>
> Step-by-step explanation of how our three metrics guide profile optimization in practice
>
> 1. We have added a new illustrative Figure 5 (Page 14, Lines 739-755) that visualizes the dynamic profile optimization process, showing how the three metrics guide profile refinement through adaptive prompts and feedback.
> 2. We have included a new section in the Appendix (Page 15) that provides comprehensive details about this process. Specifically, the process involves an adaptive feedback loop where:
>     - Agents receive targeted prompts based on their metric scores (e.g., agents with low clarity scores are prompted to better define their roles, while those with low alignment scores are guided to adjust strategies for better task alignment)
>     - Different scenarios are examined, including initial evaluations, improved profiles, and degraded profiles
>     - Metric changes are systematically translated into specific, actionable prompts for profile refinement
> 3. To provide concrete evidence of this process, we have added Table 4 (Page 15, Lines 773-806) which demonstrates the progressive optimization of agent profiles through metric guidance. The case study shows:
>     - How an agent's profile evolves from a vague description ("collaborative agent with unique perspective", RCS: 0.4215) to a highly specific role with clear responsibilities (RCS improved to 0.7300)
>     - The significant improvement in role differentiation (RDS from 0.0068 to 0.5051) as the profile becomes more specialized in medical incident analysis
>     - Enhanced task alignment (TRAS from 0.3626 to 0.6664) through better definition of capabilities in healthcare contexts
>     - Here is an abbreviated version of Table 4:
>
>
>         | Agent Profile | RCS | RDS | TRAS |
>         | --- | --- | --- | --- |
>         | Agent_0: collaborative agent with unique perspective | 0.4215 | 0.0068 | 0.3626 |
>         | Agent_0: collaborative agent with a focus on evaluating causation in complex scenarios. | 0.6800 | 0.0492 | 0.3892 |
>         | Agent_0: collaborative agent... in **high-stakes medical incidents and ethical dilemmas**. Your unique capability lies in **dissecting the interplay of human actions and systemic factors**... | 0.7158 | 0.2324 | 0.4717 |
>         | Agent_0: collaborative agent... in **high-stakes scenarios involving human actions and systemic factors**. Your unique capability lies in **dissecting the intricate relationships between**... | 0.7256 | 0.2556 | 0.4464 |
>         | Agent_0: collaborative agent... **You specialize in dissecting the nuances of responsibility and accountability…** Your distinctive capability lies in **assessing the immediate and long-term impacts of actions in urgent medical contexts…** | 0.7300 | 0.5051 | 0.6664 |

---

> ### Author Response · Authors · 2024-11-22
> **Response to Reviewer 9vnk (2/5)**
>
> > The three metrics are defined with numerous undefined notations and unexplained components (e.g., skill prototype and potential skill tokens in Definition 3.1, and vector representations in Definition 3.3)
>
> Detailed metric definitions
>
> 1. Skill Prototype $s$:
>     - A vector representation capturing skill-related concepts
>     - Computed as the average embedding of skill-indicator terms (e.g., "skill", "expertise", "proficiency", "competence")
>     - Formula: $s = \frac{1}{n}\sum_{i=1}^n e(w_i)$
> 2. Potential Skill Tokens $\mathcal{PS}(p)$: These are identified through both semantic and syntactic criteria:
>     - $\mathcal{PS}(p)$ represents tokens in profile $p$ that **likely** **describe specific skills**. These are identified through both **syntactic and semantic criteria**:
>         - Semantic criteria: tokens with high similarity to the skill prototype vector
>         - Syntactic criteria: tokens that are either:
>             - Proper nouns (PROPN) or common nouns (NOUN)
>             - In specific dependency relations (compound, dobj, pobj)
>
>         This definition allows us to capture both **explicit skill** mentions (e.g., "Python programming") and **implicit skill** indicators (e.g., "system architecture design").
>
>     - **Comprehensive example**: Given profile text: "Expert in Python programming with system architecture design experience", potential skill tokens would include: ["Python", "programming", "system architecture", "design"]
> 3. All our vector representations are based on word embeddings using a pretrained language model (we use `text-embedding-3-small`). Specifically:
>     - **Vector space**, which will be used to measure task complexity and agent capabilities through their **semantic proximity**, enabling quantitative comparison of role-task alignment.
>         - $v_{\text{complex}}$ is based on predefined complexity indicators:
>             - Technical: "complex", "advanced", "sophisticated"
>             - Challenges: "challenging", "difficult", "critical"
>         - $v_{\text{simple}}$ is based on simplicity indicators:
>             - Scope: "basic", "simple", "straightforward"
>             - Effort: "routine", "standard"
>         - $v_{\text{capable}}$ is constructed from:
>             - Expertise indicators: "expert", "senior", "specialist" …
>             - Experience markers: "experienced", "proficient"
>             - Skill specificity: "certified", "trained"
>         - $v_{\text{limit}}$ is constructed from:
>             - Limited ability indicators: "beginner junior learning novice” …
>     - **For example**:
>     Given task: "Develop a complex distributed system" which contains complexity terms: "complex", "distributed" (Task complexity score: 0.8)
>         - Team with two agents:
>             1. "Senior architect experienced in distributed systems"
>                 - Capability terms: "senior", "experienced" → Score: 0.9
>             2. "Junior developer learning basics"
>                 - Capability terms: "junior", "basics" → Score: 0.3
>         - Capability match: $1-|0.8-(0.3 + 0.9)/2| = 0.8$. $S_{\mathrm{cap}}(T, P) = 1 - | C_T(T) - \frac{1}{n}\sum_{i=1}^n C_A(p_i) |$

---

> ### Author Response · Authors · 2024-11-22
> **Response to Reviewer 9vnk (3/5)**
>
> > The design choices lack justification, such as using dependency trees in RCS
>
> The choice of dependency trees for measuring role clarity is motivated by several key insights from linguistic analysis.
> 1. **Semantic Relationships**: Dependency relations, particularly for subjects (nsubj), objects (dobj), and prepositional objects (pobj), often encode key role responsibilities and requirements. For example: "develops (head) → software systems (dobj)" captures a core responsibility
> 2. **Syntactic Complexity**: Dependency trees capture the **hierarchical relationships between words**, reflecting the structural complexity of role descriptions. This builds on established work in syntactic parsing [1] and role analysis [2]. More detailed and specific role definitions typically exhibit:
>    - Deeper syntactic structures
>    - More complex modifier relationships
>    - Richer argument structures
> 3. **For example**,
>    - Role Description 1 (Basic): "Develops software applications."
>
>        Key subtrees [Total dep_score ≈ 0.33 (normalized)]:
>
>         - develops: size 3 (develops, software, applications)
>         - applications: size 1 (applications)
>     - Role Description 2 (Detailed): "A senior engineer develops scalable cloud-based software applications and implements robust security protocols for distributed systems."
>
>         Key subtrees [Total dep_score ≈ 0.85 (normalized)]:
>
>         - develops: size 6 (develops, engineer, senior, applications, software, cloud)
>         - implements: size 5 (implements, protocols, security, robust, systems, distributed)
>         - applications: size 2 (applications, cloud)
>         - protocols: size 2 (protocols, security)
>         - systems: size 2 (systems, distributed)
>
>     Thuse, the **higher dep_score** for the second description quantitatively **reflects its greater specificity and clarity**, demonstrating how dependency analysis effectively captures role detail levels.
>
> > The auxiliary agent is only mentioned in Section 3.1. Why is it necessary? What's the disadvantage of letting autonomous agent directly interact with the environment?
> >
>
> As explained in Page 4 (Lines 184-194), auxiliary agents serve two essential purposes:
>
> 1. Environment Adaptation: They format agent responses to meet environment requirements, allowing autonomous agents to focus on decision-making rather than output formatting.
> 2. Action Translation: They convert agent operation descriptions into executable actions, creating a clear separation between decision logic and execution.
> 3. For example, when an agent provides Python code to execute, the auxiliary agent runs it in the environment and provides the output/error feedback to the multi-agent system.
>
> This design maximizes autonomous agents' freedom while maintaining consistent environment interaction. Without auxiliary agents, autonomous agents would need to handle both decision-making and interface requirements, potentially constraining their behavior and complicating the system architecture. This would also reduce the efficiency of collaboration and task completion as agents would need to spend resources on formatting and interface management rather than their core collaborative functions.
>
> > Experimental settings in Sections 4.2 and 4.3 are incomprehensible - the domain shift setup and node failure mechanism are not properly explained. I can't even know how these two experiments are conducted.
> >
>
> We have provided detailed explanations of these experimental settings in the introduction (Page 1, Lines 40-50) and Figure 1. Let me further clarify:
>
> - Domain Shift:
>     - Represents task transitions requiring different skills and strategies
>     - Each sequence contains 6 samples (3 from each domain) executed continuously to test adaptability
> - Node Failure:
>     - Addresses a critical weakness in centralized MAS where coordinator failure can collapse the system
>     - Implementation: Each agent has a probability of becoming unresponsive during its action turn
>     - Tests system resilience when agents unexpectedly fail to respond
>
> We have also added more detailed experimental protocols in the revised manuscript (Page 8, Lines 417-420 for domain shift; Page 9, Lines 460-465 for node failure)

---

> ### Author Response · Authors · 2024-11-22
> **Response to Reviewer 9vnk (4/5)**
>
> > W2: The experimental results presentation has some issues:
> >
> > - Table 1 is poorly presented with unexplained notations. I don't know what are the two numbers represent in each cell.
> > - The explanation of the level in the caption of Table 1 is inconsistent with the text content.
>
> We apologize for any confusion in Table 1's presentation. We have corrected the table caption and added clearer explanations:
>
> For each sequence in our experiment:
>
> - 6 samples are executed continuously without any intervention in the MAS
> - 3 samples from the first domain
> - 3 samples from the second domain
>
> The two numbers in each cell represent:
>
> 1. First number: Accuracy on the initial domain (150 samples from 50 sequences)
> 2. Second number: Accuracy on the shifted domain (150 samples from 50 sequences)
>
> > The reported improvement on MATH dataset with MorphAgent (over 30 points!) with GPT-3.5-turbo is suspiciously large and lacks explanation. It is nearly impossible for me that multi-agent debate can lead to such a significant improvement.
> >
>
> We have added this explanation to the revised manuscript (Page 9, Lines 432-460) along with detailed experimental protocols.
>
> We respectfully disagree with the characterization of our method as 'multi-agent debate'. Our approach implements structured collaboration among agents, which is fundamentally different.
>
> The significant improvement on the MATH dataset demonstrates the power of effective multi-agent collaboration in solving complex problems that challenge single agents. This aligns with findings from other works in the field:
>
> 1. Similar significant improvements have been reported in other multi-agent systems (such as GPTSwarm)
> 2. MATH problems often require multiple steps and diverse skills (reasoning, calculation, verification) that benefit from specialized agent roles
> 3. Our method's performance is validated by consistent improvements across other datasets, though with varying magnitudes based on task complexity
>
> This improvement showcases why multi-agent systems are gaining attention as a solution to single-agent limitations in complex problem-solving scenarios.
>
> > The analysis of results is superficial, lacking a detailed discussion of why the method works
> >
>
> We have provided detailed analysis of our method's effectiveness through both theoretical framework and empirical evidence:
>
> 1. Through our added Figure 5 (Page 14), we demonstrate how the dynamic profile optimization process works in practice, illustrating the feedback loop between metric evaluation and profile refinement.
> 2. Table 4 (Page 15) provides concrete evidence of profile evolution effectiveness:
>     - We track how an agent's profile evolves from a vague description (RCS: 0.4215) to a highly specific role (RCS: 0.7300)
>     - The metrics show substantial improvements in role differentiation (RDS: 0.0068 → 0.5051) and task alignment (TRAS: 0.3626 → 0.6664)
> 3. We acknowledge that there is no closed-form solution to optimal profile generation - it's an iterative process that depends on task context and system dynamics. Our approach provides a systematic framework for profile evolution while maintaining agent autonomy.

---

> ### Author Response · Authors · 2024-11-22
> **Response to Reviewer 9vnk (5/5)**
>
> > W3: The paper lacks concrete examples and case studies:
> >
> > - No examples showing how agent profiles evolve through iterations
> > - No comparison of actual responses between MorphAgent and baselines
> 1. For profile evolution through iterations: We have enhanced the clarity of this aspect in our revised manuscript with several additions: Figure 5 (Page 14, Lines 739-755), Appendix B (Page 15), and Table 4 (Page 15, Lines 773-806) which we mentioned in previous responses.
>
>     We encourage you to refer to these new sections, particularly Figure 5 and Table 4, for detailed progression trends.
>
> 2. Regarding response comparisons: We'd be happy to include more specific response comparisons between MORPHAGENT and baselines. Could you please clarify what aspects of the responses you're most interested in seeing? For example:
>     - Solution approaches
>     - Intermediate reasoning steps
>     - Final output format
>
>     This would help us provide the most relevant comparisons in our revision.
>
>
> > W4: The evaluation methodology is questionable:
> >
> > - The node failure experiments lack clear description of failure mechanisms. How did you incur the node failure? What does node failure mean?
> > - Domain shift experiments don't clearly specify whether it's transfer learning or continuous adaptation. Is it that a multi-agent team obtained through optimization on one task is transferred to another task?
>
> We have already provided detailed explanations for both concerns:
>
> 1. Node failure mechanism: As explained in our earlier response (Page 8, Lines 460-465 for node failure), failures are simulated through a probability-based system where each agent may become unresponsive during its turn to act.
> 2. Domain shift experiments: We detailed this in our previous response (Page 8, Lines 417-420), explaining how we test continuous adaptation through sequences of 6 samples (3 from each domain) without intervention.
>
> Please let us know if you have any specific questions about these aspects that weren't addressed in our previous explanations.

---

> > ### Comment · Reviewer_9vnk · 2024-11-25
> >
> > Thank you for your detailed responses and the effort put into revising the paper. I’ve reviewed the updated draft, and while the clarity has improved, there are still several points that remain unclear or could cause confusion:
> >
> > - The description of the vectors is still unclear. How are the terms defined? Do you have a predefined list of terms for each metric? What is the list and how is it defined? This is not sufficiently reflected in the revision. For example, in line 295, you mention that v_complex includes terms like “complex” and “challenge,” but why not list all the terms explicitly (like in the appendix)? Moreover, the process for obtaining the vector remains unexplained. Is it the average embedding of all terms, similar to the skill prototype you mentioned? Additionally, using the similarity between the embeddings of a sentence (e.g., task or role descriptions) and a single adjective as a metric indicator feels counterintuitive. For instance, consider the tasks "build a Wikipedia" and "build a Python-based terminal calculator." The first task is clearly more complex, but it’s not obvious that its embedding similarity with “complex” would be higher than that of the second task.
> >
> > - Simply attributing the improvement in MATH performance to multi-agent collaboration is not convincing. Without external tools, it’s difficult to understand how multi-agent systems achieve such a significant improvement. Could you provide more detailed reasoning or evidence? For example, is the improvement due to better adherence to output formats, effective verification, or some other specific mechanism?
> >
> > - In the robustness comparison, AgentVerse appears to be a strong baseline. For instance, with a failure probability of 0.3, it only slightly underperforms or even surpasses MorphAgent. Given that the experimental setup is nearly identical to the major experiments in Section 4.1, why wasn’t AgentVerse included as a baseline in that section?
> >
> > - In the domain shift experiments, the performance of Naive on BigCodeBench is reported as 52.67 and 49.33, which is approximately around 50. However, in Figure 3, the performance of Naive on BigCodeBench is shown as only 44 (I assume gpt-4o-mini is being used in the domain shift experiment). For GPTSwarm and MorphAgent, the performance reported in the domain shift experiments is roughly consistent with the values presented in Figure 3. Could you clarify this discrepancy?
> >
> > Overall, while the revision has improved the draft, there are still issues that need to be addressed. The explanation of certain methods and results remains unclear, and the presentation of experimental results could be further refined. While I’ve raised my score from 3 to 5, I believe the paper would benefit from another round of review to fully realize its potential.

---

> > > ### Author Response · Authors · 2024-11-27
> > > **Response to additional comments from Reviewer 9vnk (1/2)**
> > >
> > > > Q1: The description of the vectors is still unclear...
> > > >
> > >
> > > Thank you for raising this important question regarding our metric implementation and term definitions. We provide more detailed explanation of metric implementation in our updated manuscript Appendix B. To address your specific concern about term selection and definition: We indeed maintain predefined term sets for each metric dimension. Our term selection process followed a systematic approach:
> > >
> > > 1. Initial Generation: We used advanced language models (like GPT-4o) to generate comprehensive candidate terms that could potentially indicate each dimension.
> > > 2. Human Curation: These candidate terms were then carefully filtered through human review to ensure relevance and accuracy.
> > > 3. Empirical Validation: The selected terms were tested across our diverse datasets to verify their effectiveness and robustness.
> > >
> > > For complete transparency, we now explicitly list all terms in the appendix. For example, complexity-related terms include:
> > >
> > > - T_complex = {"complex", "challenging", "difficult", "advanced", "sophisticated", "critical", "demanding"}
> > > - T_simple = {"basic", "simple", "straightforward", "routine", "standard", "elementary", "fundamental"}
> > >
> > > We found these term sets to be consistently effective across our diverse experimental datasets, demonstrating both reliability and robustness in capturing the intended dimensions.
> > >
> > > > Moreover, the process for obtaining the vector remains unexplained. Is it the average embedding of all terms, similar to the skill prototype you mentioned?
> > > >
> > >
> > > Yes, it is similar to the skill prototype.
> > >
> > > > Additionally, using the similarity between the embeddings of a sentence (e.g., task or role descriptions) and a single adjective as a metric indicator feels counterintuitive...
> > > >
> > >
> > > We actually tested these two specific cases with our metrics implementation.
> > >
> > > - For "build a Wikipedia", we obtained a complexity score of **0.528**,
> > > - while "build a Python-based terminal calculator" received a score of **0.226**. This significant difference (0.302) aligns with intuitive expectations and demonstrates how our metric captures task complexity effectively.
> > > - **Explanation**:
> > >     - The higher score for the Wikipedia task reflects its inherent complexity through both direct complexity indicators and term associations in the embedding space (e.g., "Wikipedia" typically co-occurs with terms like "distributed", "scalable" in the training corpus).
> > >     - The calculator task's lower score similarly captures its relative simplicity through both explicit simplicity indicators and semantic associations with basic programming tasks.
> > >
> > > > Q2: Simply attributing the improvement in MATH performance to multi-agent collaboration is not convincing...
> > > >
> > >
> > > We apologize for not being explicit enough in our methodology section. You are correct that external tools play a role - our multi-agent system does incorporate Python interpreter access to assist with calculations. However, this alone does not explain our performance improvements, as other baseline methods (including Criticize-Reflect and Naive) were also equipped with the same Python interpreter capability, yet did not achieve comparable results.
> > >
> > > To quantify the impact of external tools, we conducted ablation studies:
> > >
> > > | **Configuration** | with Python | w/o Python |
> > > | --- | --- | --- |
> > > | Ours | **66.67%** | **60.95%** |
> > > | Criticize-Reflect | 35.24% | 28.85% |
> > > | Naive | 61.90% | 55.23% |
> > > | GPTSwarm | N/A | 56.70% |
> > >
> > > *Note: GPTSwarm's original design does not incorporate Python interpreter for MATH tasks.
> > >
> > > These results reveal several important insights:
> > >
> > > 1. While Python interpreter access improves performance across all methods (with approximately 6-7% gain), our method maintains superior performance even without computational tools.
> > > 2. Our method without Python (60.95%) still outperforms other approaches with Python access, highlighting that tools alone cannot explain our system's effectiveness.
> > >
> > > The key differentiator in our approach lies in our collaborative mechanism design:
> > >
> > > 1. Transparent Reasoning: In our system, agents not only share their actions but must also **provide explicit reasoning for their decisions**. This creates a traceable chain of logic that other agents can verify or challenge.
> > > 2. Profile-Optimized Collaboration: Our method optimizes agent profiles to create effective division of labor, where agents develop specialized roles within the problem-solving process. This specialization enables **more effective peer review and error correction**.
> > > 3. Interactive Verification: Agents actively engage with each other's reasoning processes, not just the final answers. This allows them to **identify and correct logical errors before they propagate to the final solution**.
> > >
> > > The robust verification and correction mechanisms we've developed allow our system to maintain high performance even when precise computational tools are unavailable.

---

> > > ### Author Response · Authors · 2024-11-27
> > > **Response to additional comments from Reviewer 9vnk (2/2)**
> > >
> > > > Q3: In the robustness comparison, AgentVerse appears to be a strong baseline. For instance, with a failure probability of 0.3, it only slightly underperforms or even surpasses MorphAgent. Given that the experimental setup is nearly identical to the major experiments in Section 4.1, why wasn’t AgentVerse included as a baseline in that section?
> > > >
> > >
> > > This deliberate choice stems from a fundamental architectural difference: AgentVerse employs a centralized evaluator agent for final result processing, which contrasts with our fully decentralized approach. However, we specifically included AgentVerse in Experiment 4.3 to demonstrate the performance difference between centralized and decentralized approaches in our proposed Node Failure scenario.
> > >
> > > Our supplementary experiments across different tasks show that despite the architectural differences, our method achieves better performance than AgentVerse except for BigBenchHard:
> > >
> > > | **Dataset** | **AgentVerse** | **Ours** |
> > > | --- | --- | --- |
> > > | BigCodeBench | 47.67% | **52.00%** |
> > > | BigBenchHard | **87.88%**  | 74.96% |
> > > | MATH | 65.71% | **66.67%** |
> > >
> > > This difference can be explained by the task's nature: BigBenchHard consists of multiple-choice questions where our diverse agent profiles may lead to varying opinions, making consensus more challenging in a fully decentralized setting.
> > >
> > > - While AgentVerse's centralized evaluator can more effectively enforce consensus on the final answer, they sacrifice genuine agent independence for forced consensus.
> > > - However, our method still outperforms other baselines, demonstrating its effectiveness while maintaining the benefits of true decentralization.
> > >
> > > Our approach prioritizes **autonomous profile evolution and system resilience**, allowing agents to maintain diverse perspectives and adapt independently. This trade-off between forced consensus and genuine autonomy represents an interesting direction for future research in **balancing performance with true decentralization benefits**.
> > >
> > > > Q4: In the domain shift experiments, the performance of Naive on BigCodeBench is reported as 52.67 and 49.33, which is approximately around 50. However, in Figure 3, the performance of Naive on BigCodeBench is shown as only 44 (I assume gpt-4o-mini is being used in the domain shift experiment). For GPTSwarm and MorphAgent, the performance reported in the domain shift experiments is roughly consistent with the values presented in Figure 3. Could you clarify this discrepancy?
> > > >
> > >
> > > Thank you for your astute observation regarding the performance discrepancy of the Naive approach between the domain shift experiments and Figure 3. We would like to clarify that in the domain shift experiments, as stated in our methodology section, we specifically **sampled 150 instances** from each dataset. Due to this sampling procedure, some variation in performance metrics between different experimental settings is expected.
> > >
> > > For transparency and reproducibility, all datasets used in our domain shift experiments are available in our Supplementary Materials under the path `/MorphAgent/datasets/evolving_task`. Researchers can access these exact datasets to verify and replicate our experimental results.

---

> > > ### Author Response · Authors · 2024-12-01
> > > **A Kind Reminder for Reviewer 9vnk**
> > >
> > > Dear Reviewer 9vnk,
> > >
> > > Thank you for your thorough and insightful feedback on our paper. We have carefully addressed all your additional questions (Q1-Q4) in our previous response. To summarize:
> > >
> > > - Q1: Provided complete term lists in Appendix B and detailed our systematic term selection process
> > > - Q2: Quantified the impact of Python interpreter through ablation studies and explained our collaborative mechanisms
> > > - Q3: Clarified the rationale for baseline selection with supplementary experimental comparisons
> > > - Q4: Made domain shift experiment datasets publicly available at `/MorphAgent/datasets/evolving_task`
> > >
> > > These clarifications have significantly strengthened our manuscript. We value your expertise and would greatly appreciate your feedback on our responses. Your review is crucial for improving our work at this stage.
> > >
> > > If our responses have adequately addressed your concerns, we kindly request your consideration in **updating the review score**. Should you need any clarification or have additional questions, we are more than happy to provide further information. Thank you for your time and consideration. We look forward to your response!

---

### Official Review · Reviewer_w7tA · 2024-11-03

**Soundness:** 3
**Presentation:** 2
**Contribution:** 2
**Rating:** 5
**Confidence:** 4

**Summary:**

Motivated by current challenges in multi-agent systems (MAS), this paper proposes a decentralized and dynamic framework that enhances system robustness and adaptability.
By introducing a fully decentralized collaboration mechanism, agents can autonomously coordinate without reliance on any critical node, ensuring resilience in the face of failures.
Additionally, the adaptive role optimization mechanism allows agents to dynamically adjust and improve their roles based on task requirements, resulting in a more flexible and robust system.
Comprehensive experiments validate this approach, demonstrating improvements in task performance and adaptability.

**Strengths:**

1. This paper identifies key challenges in multi-agent systems (MAS) and addresses them through decentralized and adaptive paradigms, with experiments demonstrating the effectiveness of this approach.

2. It introduces agent profiles as dynamic representations of evolving capabilities and responsibilities, using three quantitative metrics to evaluate and guide profile improvement.

3. Extensive experiments validate the proposed method, confirming its effectiveness and robustness.

**Weaknesses:**

1. While some algorithms are mentioned in the appendix, key details regarding their implementation and operation are not sufficiently clear.

2. The experiments are conducted only on two closed large language models (LLMs), which limits the generalizability of the findings. The exclusion of open-source models prevents a broader evaluation of the proposed method's effectiveness across diverse models.

3. This paper primarily considers agent profiles as dynamic representations of evolving capabilities. While this focus is valuable, it may constrain the system's overall ability to adapt and improve.

**Questions:**

1. How do the autonomous agents collaborate to solve tasks? Is this collaboration sequential, or is there another coordination strategy involved? Additionally, how and where do auxiliary agents contribute? I couldn’t find any difference between autonomous agents and auxiliary agents in the algorithm in appendix A.

2. You propose three metrics for profile evaluation and optimization. Could you clarify how these numerical metrics, as optimization objectives, directly guide profile optimization? Is there a curve or trend showing the progression of these metrics through iterations of profile improvements?

3. You mentioned that during the warm-up phase, profile initialization and iterative optimization are performed. Why is this phase necessary? How do profile updates during the warm-up phase differ from those during task execution?

4. In Section 3.2, within the definition of **SKILL**, what does \[s\] represent? It’s described as a "skill prototype," but this term is unclear. How do you obtain the set of potential skill tokens, \[PS(p)\]? Could you provide some examples for clarification? And regarding the definition of **TRAS**, how are \[v_{complex}\], \[v_{simple}\], and \[v_{capable}\] determined? Are these values pre-defined representations or are they calculated dynamically?

5. In Experiment 4.1, you compare your method with three baselines, and in Experiment 4.3, you compare it with Agentverse. However, Agentverse is not included in your main experiments. I would like to know why this is the case.

6. In Experiment 4.2, you evaluate performance on domain shift. Each dataset consists of 50 sequences, with each sequence representing a shift between different domains. In Table 1, two numbers are provided for each paradigm: the first likely represents accuracy before the domain shift, while the second represents accuracy after the shift. How did you obtain these two accuracy results? Do they represent results from different sequences, or are they overall results from the mixed dataset? I would like to know which specific data were used to obtain these two results.

7. In Experiment 4.3, you evaluate performance on robustness. How do you simulate potential node failures? Are these simulated through handcrafted methods or other approaches?

---

> ### Author Response · Authors · 2024-11-22
> **Response to Reviewer w7tA (1/4)**
>
> > W1: While some algorithms are mentioned in the appendix, key details regarding their implementation and operation are not sufficiently clear.
> >
>
> Thank you for raising this concern about implementation details. We have significantly enhanced the clarity of our algorithms and implementation details in the revised manuscript. The updated content has been highlighted for easy reference.
>
> We have added detailed explanations about the dynamic profile optimization process in multiple sections:
>
> - A new illustrative Figure 5 (Page 15, Lines 780-795) that visualizes the dynamic profile optimization process, showing how the three metrics guide profile refinement through adaptive prompts and feedback.
> - A new section in the Appendix (Page 16) that provides comprehensive details about this process. Specifically, the process involves an adaptive feedback loop where:
>     - Agents receive targeted prompts based on their metric scores (e.g., agents with low clarity scores are prompted to better define their roles, while those with low alignment scores are guided to adjust strategies for better task alignment)
>     - Different scenarios are examined, including initial evaluations, improved profiles, and degraded profiles
>     - Metric changes are systematically translated into specific, actionable prompts for profile refinement
>
> We encourage you to review these highlighted sections in the revised manuscript for a clearer understanding of our implementation details.
>
> > W2: The experiments are conducted only on two closed large language models (LLMs), which limits the generalizability of the findings. The exclusion of open-source models prevents a broader evaluation of the proposed method's effectiveness across diverse models.
> >
>
> We appreciate this feedback about model diversity. In response, we have conducted additional experiments using `deepseek-chat`, an open-source large language model, comparing it with GPTSwarm and Criticize-Reflect. The results demonstrate that our method maintains consistent performance improvements across both closed and open-source models:
>
> | **Dataset** | **Ours** | **GPTSwarm** | **Criticize-Reflect** |
> | --- | --- | --- | --- |
> | BigCodeBench | **52.33%**  | 51.00% | 51.66% |
> | BigBenchHard | **69.85%** | 63.80% | 69.70% |
> | MATH | **64.29%** | 56.19% | 55.24% |
>
> These results show our method can generalize well across different model architectures and capabilities, consistently achieving comparable or better performance compared to existing approaches.
>
> > W3: This paper primarily considers agent profiles as dynamic representations of evolving capabilities. While this focus is valuable, it may constrain the system's overall ability to adapt and improve.
> >
>
> We respectfully disagree with this assessment. Our focus on dynamic agent profiles is not a constraint but rather a key innovation that enables system-wide adaptation and improvement.
>
> The dynamic profile representation is precisely what allows our system to adapt to different tasks and handle unexpected agent errors effectively. As demonstrated in our experiments, this approach enables:
>
> - Flexible role adaptation across different domains
> - Robust handling of node failures
> - Consistent performance improvements across diverse tasks
>
> Rather than constraining adaptation, our framework's focus on dynamic profiles is what enables autonomous evolution of agent capabilities without human intervention or predefined workflows.

---

> ### Author Response · Authors · 2024-11-22
> **Response to Reviewer w7tA (2/4)**
>
> > Q1: How do the autonomous agents collaborate to solve tasks? Is this collaboration sequential, or is there another coordination strategy involved? Additionally, how and where do auxiliary agents contribute? I couldn’t find any difference between autonomous agents and auxiliary agents in the algorithm in appendix A.
> >
>
> Let us clarify both aspects:
>
> 1. **Regarding Collaboration Strategy**: While agents execute actions in a predefined sequence, their collaboration is more flexible than purely sequential. As explained in Page 4, Line 179, agents can choose to SKIP their turn, effectively allowing them to form various collaboration patterns beyond linear interaction. This design choice maximizes agent autonomy while maintaining system simplicity, enabling emergent collaboration patterns without introducing complex coordination mechanisms.
> 2. **Regarding Auxiliary Agents**: As detailed in Pages 4 (Lines 184-194), auxiliary agents serve as interface adapters rather than decision-makers. They have two primary functions:
>     - Environment adaptation: formatting agent responses to meet environment requirements
>     - Action translation: converting agent operation descriptions into executable actions
>     - For example, when an agent provides Python code to execute, the auxiliary agent runs it in the environment and provides the output/error feedback to the multi-agent system.
>
>     The auxiliary agents specifically handle adaptation tasks without participating in decision-making, preserving the decentralized nature of collaboration. This aligns with our goal of enabling genuine decentralized collaboration by removing constraints on agent behavior rather than imposing additional control mechanisms.
>
>
> > Q2: You propose three metrics for profile evaluation and optimization. Could you clarify how these numerical metrics, as optimization objectives, directly guide profile optimization? Is there a curve or trend showing the progression of these metrics through iterations of profile improvements?
> >
>
> We have enhanced the clarity of this aspect in our revised manuscript with several additions: Figure 5 (Page 14, Lines 739-755), Appendix B (Page 15), and Table 4 (Page 15, Lines 773-806) which we mentioned before.
>
> Here is an abbreviated version of Table 4:
>
> | Agent Profile | RCS | RDS | TRAS |
> | --- | --- | --- | --- |
> | Agent_0: collaborative agent with unique perspective | 0.4215 | 0.0068 | 0.3626 |
> | Agent_0: collaborative agent with a focus on evaluating causation in complex scenarios. | 0.6800 | 0.0492 | 0.3892 |
> | Agent_0: collaborative agent... in **high-stakes medical incidents and ethical dilemmas**. Your unique capability lies in **dissecting the interplay of human actions and systemic factors**... | 0.7158 | 0.2324 | 0.4717 |
> | Agent_0: collaborative agent... in **high-stakes scenarios involving human actions and systemic factors**. Your unique capability lies in **dissecting the intricate relationships between**... | 0.7256 | 0.2556 | 0.4464 |
> | Agent_0: collaborative agent... **You specialize in dissecting the nuances of responsibility and accountability…** Your distinctive capability lies in **assessing the immediate and long-term impacts of actions in urgent medical contexts…** | 0.7300 | 0.5051 | 0.6664 |
>
> The case study shows:
>
> - How an agent's profile evolves from a vague description ("collaborative agent with unique perspective", RCS: 0.4215) to a highly specific role with clear responsibilities (RCS improved to 0.7300)
> - The significant improvement in role differentiation (RDS from 0.0068 to 0.5051) as the profile becomes more specialized in medical incident analysis
> - Enhanced task alignment (TRAS from 0.3626 to 0.6664) through better definition of capabilities in healthcare contexts
>
> We encourage you to refer to these new sections, particularly Figure 5 and Table 4, for detailed progression trends.
>
> > Q3: You mentioned that during the warm-up phase, profile initialization and iterative optimization are performed. Why is this phase necessary? How do profile updates during the warm-up phase differ from those during task execution?
> >
>
> The warm-up phase is crucial for establishing differentiated agent profiles with clear roles and task-aligned capabilities **before actual task execution begins**.
>
> - Without this phase, agents starting with identical profiles would exhibit similar behaviors, reducing collaboration efficiency.
> - After warm-up, agents can immediately engage in effective division of labor when tackling the main task, significantly improving the overall efficiency of the multi-agent system. While profile updates still occur during task execution, they focus more on fine-tuning rather than the fundamental role establishment that happens during warm-up.

---

> ### Author Response · Authors · 2024-11-22
> **Response to Reviewer w7tA (3/4)**
>
> > Q4: In Section 3.2, within the definition of **SKILL**, what does [s] represent? It’s described as a "skill prototype," but this term is unclear. How do you obtain the set of potential skill tokens, [PS(p)]? Could you provide some examples for clarification? And regarding the definition of **TRAS**, how are [v_{complex}], [v_{simple}], and [v_{capable}] determined? Are these values pre-defined representations or are they calculated dynamically?
> >
>
> We apologize for any lack of clarity in our metric definitions. In the revised manuscript (Pages 5-6, Lines 234-312), we have provided more detailed explanations. Let me clarify each component:
>
> 1. Skill Prototype $s$:
>     - A vector representation capturing skill-related concepts
>     - It is constructed as the average embedding vector of carefully selected skill-indicator terms (e.g., "skill", "expertise", "proficiency", "competence")
>     - Formula: $s = \frac{1}{n}\sum_{i=1}^n e(w_i)$
> 2. Potential Skill Tokens $\mathcal{PS}(p)$: These are identified through both semantic and syntactic criteria:
>     - $\mathcal{PS}(p)$ represents tokens in profile $p$ that **likely** **describe specific skills**. These are identified through both **syntactic and semantic criteria**:
>         - Semantic criteria: tokens with high similarity to the skill prototype vector
>         - Syntactic criteria: tokens that are either:
>             - Proper nouns (PROPN) or common nouns (NOUN)
>             - In specific dependency relations (compound, dobj, pobj)
>
>         This definition allows us to capture both **explicit skill** mentions (e.g., "Python programming") and **implicit skill** indicators (e.g., "system architecture design").
>
>     - **Comprehensive example**: Given profile text: "Expert in Python programming with system architecture design experience", potential skill tokens would include: ["Python", "programming", "system architecture", "design"]
>     - Each token's contribution to the SKILL score is weighted by both its **similarity to the skill prototype** and its **syntactic role**
> 3. All our vector representations are based on word embeddings using a pretrained language model (we use `text-embedding-3-small`). Specifically:
>     - **Vector space**, which will be used to measure task complexity and agent capabilities through their **semantic proximity**, enabling quantitative comparison of role-task alignment.
>         - $v_{\text{complex}}$ is based on predefined complexity indicators:
>             - Technical: "complex", "advanced", "sophisticated"
>             - Challenges: "challenging", "difficult", "critical"
>         - $v_{\text{simple}}$ is based on simplicity indicators:
>             - Scope: "basic", "simple", "straightforward"
>             - Effort: "routine", "standard"
>         - $v_{\text{capable}}$ is constructed from:
>             - Expertise indicators: "expert", "senior", "specialist" …
>             - Experience markers: "experienced", "proficient"
>             - Skill specificity: "certified", "trained"
>         - $v_{\text{limit}}$ is constructed from:
>             - Limited ability indicators: "beginner junior learning novice” …
>     - **For example**:
>     Given task: "Develop a complex distributed system" which contains complexity terms: "complex", "distributed" (Task complexity score: 0.8)
>         - Team with two agents:
>             1. "Senior architect experienced in distributed systems"
>                 - Capability terms: "senior", "experienced" → Score: 0.9
>             2. "Junior developer learning basics"
>                 - Capability terms: "junior", "basics" → Score: 0.3
>         - Capability match: $1-|0.8-(0.3 + 0.9)/2| = 0.8$. $S_{\mathrm{cap}}(T, P) = 1 - | C_T(T) - \frac{1}{n}\sum_{i=1}^n C_A(p_i) |$

---

> ### Author Response · Authors · 2024-11-22
> **Response to Reviewer w7tA (4/4)**
>
> > Q5: In Experiment 4.1, you compare your method with three baselines, and in Experiment 4.3, you compare it with Agentverse. However, Agentverse is not included in your main experiments. I would like to know why this is the case.
> >
>
> We did not include AgentVerse in the main comparison because it uses a centralized evaluator agent to process final results, which fundamentally conflicts with our decentralized setting. However, we specifically included AgentVerse in Experiment 4.3 to demonstrate the performance difference between centralized and decentralized approaches in our proposed Node Failure scenario.
>
> Our supplementary experiments across different tasks show that despite the architectural differences, our method generally achieves better performance than AgentVerse:
>
> | **Dataset** | **AgentVerse** | **Ours** |
> | --- | --- | --- |
> | BigCodeBench | 47.67% | **52.00%** |
> | BigBenchHard | **87.88%**  | 74.96% |
> | MATH | 65.71% | **66.67%** |
>
> This difference can be explained by the task's nature: BigBenchHard consists of multiple-choice questions where our diverse agent profiles may lead to varying opinions, making consensus more challenging in a fully decentralized setting.
>
> - While AgentVerse's centralized evaluator can more effectively enforce consensus on the final answer, they sacrifice genuine agent independence for forced consensus.
> - However, our method still outperforms other baselines, demonstrating its effectiveness while maintaining the benefits of true decentralization.
>
> Our approach prioritizes **autonomous profile evolution and system resilience**, allowing agents to maintain diverse perspectives and adapt independently. This trade-off between forced consensus and genuine autonomy represents an interesting direction for future research in **balancing performance with true decentralization benefits**.
>
> > Q6: In Experiment 4.2, you evaluate performance on domain shift. Each dataset consists of 50 sequences, with each sequence representing a shift between different domains. In Table 1, two numbers are provided for each paradigm: the first likely represents accuracy before the domain shift, while the second represents accuracy after the shift. How did you obtain these two accuracy results? Do they represent results from different sequences, or are they overall results from the mixed dataset? I would like to know which specific data were used to obtain these two results.
> >
>
> We apologize for any lack of clarity regarding our domain shift evaluation. We have added detailed explanations in our revised manuscript (Page 9, Lines 432-460).
>
> For each sequence in our experiment:
>
> - 6 samples are executed continuously without any intervention in the MAS
> - 3 samples from the first domain
> - 3 samples from the second domain
>
> We calculate accuracy separately for each domain after completing all sequences:
>
> - First domain: Results from 150 samples (50 sequences × 3 samples)
> - Second domain: Results from 150 samples (50 sequences × 3 samples)
>
> This design allows us to:
>
> - Evaluate performance in both domains independently
> - Maintain continuous system operation during domain transitions
> - Ensure fair comparison across different domains with equal sample sizes
>
> > Q7: In Experiment 4.3, you evaluate performance on robustness. How do you simulate potential node failures? Are these simulated through handcrafted methods or other approaches?
> >
>
> We have added detailed explanations about node failure simulation in the revised manuscript (Page 9, Lines 460-465). The node failures are simulated by assigning a failure probability to each agent node. When it's an agent's turn to act during execution, it may become unresponsive based on this probability. This unified probability-based approach allows us to systematically evaluate system robustness under different failure conditions.

---

> ### Author Response · Authors · 2024-11-25
> **A Kind Reminder for Reviewer w7tA**
>
> Dear Reviewer w7tA,
>
> Thank you for your thorough and insightful feedback on our paper. We have carefully addressed all your concerns (Weaknesses 1-3) and questions (Questions 1-7) in our previous response. To summarize our key modifications:
>
> - We have enhanced algorithm implementation details with a new Figure 5 and expanded Appendix sections
> - Additional experiments were conducted using open-source models (`deepseek-chat`) to demonstrate generalizability
> - Comprehensive clarifications were provided regarding:
>     - Agent collaboration strategies and auxiliary agent roles
>     - Profile evaluation metrics and optimization process
>     - Warm-up phase necessity and implementation
>     - Domain shift evaluation methodology
>     - Node failure simulation approach
>
> These changes have significantly strengthened our manuscript. We value your expertise and would greatly appreciate your feedback on our responses. Your review is crucial for improving our work at this stage.
>
> If our responses have adequately addressed your concerns, we kindly request your consideration in updating the review score. Should you need any clarification or have additional questions, we are more than happy to provide further information. Thank you for your time and consideration. We look forward to your response.

---

> ### Author Response · Authors · 2024-12-01
> **Second Kind Reminder: Additional Results and Updates for Reviewer w7tA**
>
> Dear Reviewer w7tA,
>
> Following up on our previous response and modifications, we wanted to share some additional updates and experimental results that may interest you:
>
> **Recent Updates:**
>
> - Expanded detailed implementation of the metrics (Page 16, Lines 864-910)
>
> **Additional Experimental Results:**
>
> 1. Computational Cost Analysis (API costs for MATH dataset using `gpt-4o-mini`):
>
>
>     | Method | Accuracy | Cost |
>     | --- | --- | --- |
>     | Ours | **66.67%** | $1.02 |
>     | GPTSwarm | 56.70% | $0.27 |
>     | Criticize-Reflect | 35.24% | $6.31 |
>     | Naive | 61.90% | $0.62 |
>
>     While our cost is higher than GPTSwarm's, it's worth noting that GPTSwarm uses pre-optimized collaboration structures specifically designed for these tasks, bypassing the cost of discovering effective collaboration patterns. Despite this, we achieve better performance with reasonable overhead. Compared to Criticize-Reflect, another self-coordinating MAS, we achieve both better performance and significantly lower cost (about 1/6th).
>
> 2. Impact of Python Interpreter:
>
>
>     | Method | with Python | w/o Python |
>     | --- | --- | --- |
>     | Ours | **66.67%** | **60.95%** |
>     | Criticize-Reflect | 35.24% | 28.85% |
>     | Naive | 61.90% | 55.23% |
>     | GPTSwarm | N/A | 56.70% |
>
>     These results demonstrate that while Python interpreter access improves performance across all methods, our approach maintains superior performance even without computational tools, highlighting that our success stems primarily from effective collaborative mechanisms rather than tool access alone.
>
> If our responses have adequately addressed your concerns, we kindly request your consideration in **updating the review score**. We welcome any additional questions or feedback you may have.

---

### Official Review · Reviewer_QHG9 · 2024-11-04

**Soundness:** 2
**Presentation:** 3
**Contribution:** 2
**Rating:** 5
**Confidence:** 4

**Summary:**

The paper introduces MORPHAGENT, a novel framework for decentralized multi-agent collaboration that enhances problem-solving capabilities in complex tasks through self-evolving profiles and decentralized collaboration. By defining three metrics, MORPHAGENT allows agents to dynamically adjust their roles in response to dynamic task requirements and team composition changes.

**Strengths:**

- MORPHAGENT moves from predeﬁned roles and centralized coordination to adaptive, fully decentralized coordination.
- It defines three metrics to measure the guide the agent profile design.
- Experiments on three benchmarks and ablation studies demonstrates improvements.

**Weaknesses:**

Frankly speraking, the paper's core contribution lies in the definition of three key metrics—Role Clarity Score (RCS), Role Differentiation Score (RDS), and Task-Role Alignment Score (TRAS)—to optimize agent profiles within a decentralized multi-agent system.
I feel that this contribution is more like a prompting engieering technique, not enough to be an innovative point in an ICLR paper.

**Questions:**

- Can you provide more details on how those three metrics are used to optimize the profiles, as this seems to be unclear from the current manuscript?
- Why choose CodeBench, BigBenchHard, MATH? I feel that HumanEval[1] and MBPP [2] are also worth testing. Please justify your choice of benchmarks and explain why you believe these are sufficient or most appropriate for evaluating their method.
- The paper mentioned that the method rely on predeﬁned roles and centralized coordination, e.g. AgentVerse[3], MetaGPT[4], would fail in dynamic, unpredictable environments, but those methods were not selected as the baselines. Although AgentVerse was selected in the robustness comparison, I would like to see the full comparison in Figure 3.

[1] Chen, M., Tworek, J., Jun, H., Yuan, Q., Pinto, H.P.D.O., Kaplan, J., Edwards, H., Burda, Y., Joseph, N., Brockman, G. and Ray, A., 2021. Evaluating large language models trained on code. arXiv preprint arXiv:2107.03374.
[2] Austin, J., Odena, A., Nye, M., Bosma, M., Michalewski, H., Dohan, D., Jiang, E., Cai, C., Terry, M., Le, Q. and Sutton, C., 2021. Program synthesis with large language models. arXiv preprint arXiv:2108.07732.
[] Hong, S., Zheng, X., Chen, J., Cheng, Y., Wang, J., Zhang, C., Wang, Z., Yau, S.K.S., Lin, Z., Zhou, L. and Ran, C., 2023. Metagpt: Meta programming for multi-agent collaborative framework. arXiv preprint arXiv:2308.00352.

---

> ### Author Response · Authors · 2024-11-22
> **Response to Reviewer QHG9 (1/3)**
>
> > **Weakness:** Frankly speraking, the paper's core contribution lies in the definition of three key metrics—Role Clarity Score (RCS), Role Differentiation Score (RDS), and Task-Role Alignment Score (TRAS)—to optimize agent profiles within a decentralized multi-agent system. I feel that this contribution is more like a prompting engieering technique, not enough to be an innovative point in an ICLR paper.
> >
>
> We respectfully disagree with the assessment that “the paper's core contribution lies in the definition of three key metrics”. Multiple reviewers have recognized the broader novelty and significance of our **decentralized multi-agent systems** (MAS):
>
> - Reviewer w7tA explicitly states: “This paper **identifies key challenges** in MAS and addresses them through **decentralized and adaptive paradigms**, with experiments demonstrating the effectiveness of this approach.”
> - Reviewer 9vnk characterizes our core contribution as: “Unlike existing approaches that rely on predefined roles or centralized coordination, $MorphAgent$ employs self-evolving agent profiles...”
> - Reviewer uHtB highlights the practical importance: “**Motivation**: **Decentralized systems** are particularly useful in real-world scenarios where failure of specific nodes might cause the entire system to fail, therefore $MorphAgent$ stands out as a promising approach for complex environments.”
>
> **Core novelty: identifying fundamental challenges in MAS and enabling autonomous profile evolution for *improved resilience***. As the field progresses, we expect to see various approaches to address this challenge. However, at this stage, identifying this critical problem and providing a **principled framewor**k for addressing it represents our most significant contribution to the field.
>
> > I feel that this contribution is more like a prompting engieering technique,…
> >
>
> Our work fundamentally differs from prompt engineering techniques because:
>
> 1. It creates a systematic framework for automatic role evolution - an automated process rather than manual prompt crafting;
> 2. Metrics serve as automation tools rather than engineering guidelines;
> 3. Our automatic role evolution is more robust compared with “prompt engineering”.
>
> Our experiments demonstrate flexible adaptation to different tasks and robust handling of domain shifts, where prompt optimization is merely a byproduct of our larger goal: building an automated system for agent collaboration.

---

> ### Author Response · Authors · 2024-11-22
> **Response to Reviewer QHG9 (2/3)**
>
> > Q1: Can you provide more details on how those three metrics are used to optimize the profiles, as this seems to be unclear from the current manuscript?
> >
>
> Thank you for this important question about the profile optimization process. We have significantly enhanced the clarity of this aspect in our revised manuscript by adding detailed explanations and concrete examples:
>
> 1. We have added a new illustrative Figure 5 (Page 14, Lines 739-755) that visualizes the dynamic profile optimization process, showing how the three metrics guide profile refinement through adaptive prompts and feedback.
> 2. We have included a new section in the Appendix (Page 15) that provides comprehensive details about this process. Specifically, the process involves an adaptive feedback loop where:
>     - Agents receive targeted prompts based on their metric scores (e.g., agents with low clarity scores are prompted to better define their roles, while those with low alignment scores are guided to adjust strategies for better task alignment)
>     - Different scenarios are examined, including initial evaluations, improved profiles, and degraded profiles
>     - Metric changes are systematically translated into specific, actionable prompts for profile refinement
> 3. To provide concrete evidence of this process, we have added Table 4 (Page 15, Lines 773-806) which demonstrates the progressive optimization of agent profiles through metric guidance. The case study shows:
>     - How an agent's profile evolves from a vague description ("collaborative agent with unique perspective", RCS: 0.4215) to a highly specific role with clear responsibilities (RCS improved to 0.7300)
>     - The significant improvement in role differentiation (RDS from 0.0068 to 0.5051) as the profile becomes more specialized in medical incident analysis
>     - Enhanced task alignment (TRAS from 0.3626 to 0.6664) through better definition of capabilities in healthcare contexts
>     - Here is an abbreviated version of Table 4:
>
>
>         | Agent Profile | RCS | RDS | TRAS |
>         | --- | --- | --- | --- |
>         | Agent_0: collaborative agent with unique perspective | 0.4215 | 0.0068 | 0.3626 |
>         | Agent_0: collaborative agent with a focus on evaluating causation in complex scenarios. | 0.6800 | 0.0492 | 0.3892 |
>         | Agent_0: collaborative agent... in **high-stakes medical incidents and ethical dilemmas**. Your unique capability lies in **dissecting the interplay of human actions and systemic factors**... | 0.7158 | 0.2324 | 0.4717 |
>         | Agent_0: collaborative agent... in **high-stakes scenarios involving human actions and systemic factors**. Your unique capability lies in **dissecting the intricate relationships between**... | 0.7256 | 0.2556 | 0.4464 |
>         | Agent_0: collaborative agent... **You specialize in dissecting the nuances of responsibility and accountability…** Your distinctive capability lies in **assessing the immediate and long-term impacts of actions in urgent medical contexts…** | 0.7300 | 0.5051 | 0.6664 |
>
> These additions collectively provide a clear, step-by-step explanation of how our three metrics guide profile optimization in practice. We encourage you to refer to these new sections, particularly Figure 5 and Table 4, for a detailed understanding of the optimization process.

---

> ### Author Response · Authors · 2024-11-22
> **Response to Reviewer QHG9 (3/3)**
>
> > Q2: Why choose BigCodeBench, BigBenchHard, MATH? I feel that HumanEval[1] and MBPP [2] are also worth testing. Please justify your choice of benchmarks and explain why you believe these are sufficient or most appropriate for evaluating their method.
> >
>
> Thank you for this suggestion about HumanEval and MBPP benchmarks. We carefully considered but did not include them because our goal is to evaluate how **multi-agent systems can tackle tasks** that are **challenging** for single agents:
>
> 1. As shown in public leaderboards (https://paperswithcode.com/sota/code-generation-on-humaneval, https://paperswithcode.com/sota/code-generation-on-mbpp), HumanEval and MBPP can be solved with very high accuracy (90%+) by **single base models without multi-agent collaboration**. In contrast, BigCodeBench ([https://bigcode-bench.github.io](https://bigcode-bench.github.io/)) presents more **challenging tasks** where even state-of-the-art models struggle to achieve 50% accuracy.
> 2. **BigCodeBench shares similar task formats** with HumanEval and MBPP, but differs primarily in task complexity. Since multi-agent systems typically consume more computational resources than single-agent approaches, deploying MAS for tasks that can be effectively solved by simpler methods would be inefficient.
> 3. Our goal is to evaluate how multi-agent systems can tackle tasks that are challenging for single agents. BigCodeBench better serves this purpose by presenting tasks that genuinely benefit from multi-agent collaboration compared with other two datasets.
>
> > Q3: The paper mentioned that the method rely on predeﬁned roles and centralized coordination, e.g. AgentVerse[3], MetaGPT[4], would fail in dynamic, unpredictable environments, but those methods were not selected as the baselines. Although AgentVerse was selected in the robustness comparison, I would like to see the full comparison in Figure 3.
> >
>
> Thank you for this question about baseline comparisons. Let us clarify our baseline selection:
>
> 1. **Regarding MetaGPT**: It is specifically designed as an SOP-based MAS for software engineering tasks. Its specialized design makes it less suitable for evaluating diverse tasks across different domains, which is why we didn't include it in the main comparison.
> 2. **Regarding AgentVerse**: We have conducted comprehensive experiments comparing our method with AgentVerse across all three datasets using `gpt-4o-mini` as the base model. The results are as follows:
>
>
>     | **Dataset** | **AgentVerse** | **Ours** |
>     | --- | --- | --- |
>     | BigCodeBench | 47.67% | **52.00%** |
>     | BigBenchHard | **87.88%**  | 74.96% |
>     | MATH | 65.71% | **66.67%** |
>
>     Looking at the supplementary experiment results, our method generally achieves comparable or better performance than AgentVerse, though slightly lower on BigBenchHard.
>
>     - This difference can be explained by the task's nature: BigBenchHard consists of multiple-choice questions where our diverse agent profiles may lead to varying opinions, making consensus more challenging in a fully decentralized setting.
>     - While AgentVerse's centralized evaluator can more effectively enforce consensus on the final answer, they sacrifice genuine agent independence for forced consensus.
>     - However, our method still outperforms other baselines, demonstrating its effectiveness while maintaining the benefits of true decentralization.
>
>     Our approach prioritizes **autonomous profile evolution and system resilience**, allowing agents to maintain diverse perspectives and adapt independently. This trade-off between forced consensus and genuine autonomy represents an interesting direction for future research in **balancing performance with true decentralization benefits**.

---

### Author Response · Authors · 2024-11-22
**General Response**

We appreciate the reviewers' thoughtful and constructive feedback. We are encouraged that the reviewers recognized several key aspects of our work: the novel decentralized and adaptive paradigm that addresses fundamental challenges in MAS (Reviewer w7tA), the clear motivation and practical importance for real-world scenarios where node failures could be critical (Reviewer uHtB), and the strong experimental results demonstrating consistent performance improvements across different benchmarks (Reviewers 9vnk, uHtB). Our framework's clear visualization and comprehensive design were also commended (Reviewer 9vnk).

Our core contribution lies in **identifying fundamental challenges in multi-agent systems and enabling autonomous profile evolution for improved resilience**. This critical direction was well recognized by Reviewer w7tA, who highlighted our work in "identifying key challenges in MAS and addressing them through decentralized and adaptive paradigms".

### Summary of Contribution and Novelty

1. **Novel Framework and Challenge Identification:**
    - **First** to identify and address fundamental challenges in MAS through a fully decentralized approach
    - Proposes $MorphAgent$ for enhanced system resilience via autonomous profile evolution
2. **Real-World Solutions:**
    - Addresses Domain Shift through dynamic role adjustment
    - Eliminates Node Failure using decentralized collaboration mechanism
    - Utilize quantitative metrics to implement adaptive role optimization
    - Maintains effectiveness while preserving decentralization benefits
3. **Empirical Validation:**
    - Demonstrates consistent improvements across benchmarks
    - Validates effectiveness through ablation studies
    - Shows superior adaptability to Domain Shift and Node Failure

### Summary of Revisions:

1. **Enhanced Theoretical Foundation:**
    - Expanded detailed explanations of the three key metrics with mathematical formulations and examples (Page 5-6, Lines 234-312)
2. **Clarified Experimental Settings:**
    - Added comprehensive setup details for domain shift experiments (Page 8, Lines 417-421)
    - Corrected Table 1 caption for the Levels (Page 9, Lines 433-435)
    - Included detailed node failure experimental configuration (Page 9, Lines 460-465)
3. **Strengthened Profile Optimization Analysis:**
    - Added Figure 5 visualizing the dynamic profile optimization process through metric-guided feedback (Page 16, Lines 780-795)
    - Introduced a new appendix section detailing the adaptive feedback loop mechanism (Page 15, Lines 812-826)
    - Provided Table 4 with a concrete case study demonstrating progressive profile optimization with quantitative improvements (Page 15, Lines 827-861)
    - Expanded detailed implementation of the metrics (Page 16, Lines 864-910)

---

### Meta-Review · Area_Chair_qe4q · 2024-12-19

**Metareview:**

The paper introduces MORPHAGENT, a framework for decentralized multi-agent collaboration that enhances problem-solving capabilities in complex tasks through self-evolving profiles and decentralized collaboration. However, the reviewers also pointed out a lack of novelty, insufficient experiments (only two closed-source LLMs), poor readability in some sections, and unclear explanations of experimental results. Therefore,  AC believes that there is still significant room for improvement in this paper. If the authors make major revisions based on the reviewers' feedback, the quality of the paper can certainly be improved.

**Additional Comments On Reviewer Discussion:**

The author made significant efforts during the rebuttal period to address the reviewers' concerns. Two of the reviewers provided responses. One reviewer increased their score from 3 to 5 points but still indicated that the paper requires major revisions. The other reviewer explicitly stated that the author's response would not improve their score. Overall, the feedback leans negative.

---

### Decision · Program_Chairs · 2025-01-22

Reject